# The Eocene-Oligocene Transition in the Paratethys: Boreal Water Ingression and its Paleoceanographic Implications

Mustafa Yücel Kaya[1*], Henk Brinkhuis[2, 3], Chiara Fioroni[4], Serdar Görkem Atasoy[1], Alexis Licht[5], Dirk Nürnberg[6], Taylan Vural[1]

[1]Department of Geological Engineering, Middle East Technical University, Ankara, 06800, Türkiye
[2]Oceans Systems research (OCS), NIOZ Royal Netherlands Institute of Sea Research, Texel, 1790 AB The Netherlands
[3]Earth Sciences dept., Laboratory of Palaeobotany and Palynology, Faculty of Geosciences, Utrecht University, 3584 CB Utrecht, The Netherlands
[4]Dipartimento di Scienze Chimiche e Geologiche, Università degli Studi di Modena e Reggio Emilia, Modena, 41121, Italy
[5]Aix-Marseille Université, CNRS, IRD, INRAE, Collège de France, CEREGE, Technopôle de l'Arbois-Méditerranée, BP80, 13545 Aix-en-Provence, France
[6]GEOMAR Helmholtz Centre for Ocean Research Kiel, Kiel, D-24148, Germany
[*]Present address: Geological Institute, RWTH Aachen University, Germany

*Correspondence to*: Mustafa Y. Kaya (mustafaky@gmail.com)

**Abstract.** The Eocene-Oligocene Transition (EOT) represents a pivotal period in Earth's climatic history, marked by the onset of Antarctic glaciation and global cooling. While deep-sea records have extensively documented this transition, its impacts on marginal and epicontinental seas remain less understood. This study investigates the impacts of the EOT in the Karaburun composite section, located in the eastern Paratethys. Using a multidisciplinary approach that integrates biostratigraphy, geochemistry, geochronology and sequence stratigraphy, a robust chronostratigraphic framework for the latest Eocene to early Oligocene was established. The stable isotopic shifts observed in benthic and planktic foraminifera $\delta^{18}O$ and $\delta^{13}C$ records at Karaburun align with global patterns but also reveal regional effects, such as freshwater influx and basin restriction, specific to the semi-restricted Paratethys. The abrupt negative $\delta^{18}O$ shift across the EOB in the Paratethys reflects boreal water ingressions driven by the onset of anti-estuarine circulation between the Nordic seas and Atlantic and the closure of the Arctic-Atlantic gateway, which redirected cold, low-salinity boreal waters through interconnected basins towards the Paratethys. These findings highlight the interplay between global climate drivers and regional hydrological dynamics, providing critical insights into the evolution of marginal marine environments during the EOT. Our results underscore the significance of the Paratethys as a unique archive for studying the onset of global icehouse climate conditions and regional responses.

## 1 Introduction

The Earth's geological history has witnessed several significant long-term climate transitions, along with short-term disruptions to the carbon cycle. The most recent of these transitions occurred over the last 50 million years during the Cenozoic and is characterized by a long-term cooling trend and a decline in atmospheric $CO_2$ levels, culminating in the onset of Antarctic glaciation during the Eocene-Oligocene Transition (EOT) (e.g., Zachos et al., 2001; Caves et al., 2016). The EOT marks the

end of an extended period of predominantly greenhouse conditions and represents a phase of accelerated biotic change lasting
approximately 500–800 kyr, bracketing the Eocene-Oligocene Boundary (EOB) (Coxall & Pearson, 2007; Hutchinson et al.,
2021). This transition is also associated with a deepening of the ocean's calcite compensation depth (CCD) (Coxall et al.,
2005), a northward migration of the Intertropical Convergence Zone (ITCZ) (Hyeong et al., 2016), and increased seasonality
in northern high latitudes (Eldrett et al., 2009).
The Antarctic glaciation events during the Oligocene are referred to as Oi events (Pälike et al., 2006; Pekar and Miller, 1996).
At deep-ocean sites Oi glaciation events are characterized by positive excursions in the oxygen isotope records of benthic
foraminifera (Miller et al., 1991; Pälike et al., 2006; Wade and Pälike, 2004). During the earliest Oligocene there are two
significant cooling/glaciation events: (1) Oi-1 (Early Oligocene Glacial Maximum, EOGM  of Hutchinson et al., 2020) and
(2) Oi-1a, corresponding to early part of paleomagnetic Subchron C12r, took place during the early Rupelian (Pekar et al.,
2002 and references therein). The EOGM, spanning 490 kyr from ~33.65 to 33.16 Ma, marks a sustained period of cold climate
and glaciation during paleomagnetic Subchron C13n (Hutchinson et al., 2020). The Oi-1a glaciation event corresponds to the
appearance of the cold-water dinocyst taxa *Svalbardella cooksoniae* and/or *Svalbardella* spp. in the North Sea Basin
(Śliwińska, 2019a and references therein). Episodes of southward migration of *Svalbardella cooksoniae* have consequently
been interpreted as evidence of cooling events (e.g., Van Simaeys et al., 2005). *Svalbardella cooksoniae* has also been
documented in a brief interval during the earliest Oligocene at numerous sites across the Northern Atlantic and Western
Tethyan regions (Śliwińska and Heilmann-Clausen, 2011).

The most comprehensive insights into the EOT come from deep-sea marine records of benthic foraminiferal oxygen and carbon
isotopes ($\delta^{18}O$ and $\delta^{13}C$), extensively analyzed using cores from the Deep Sea Drilling Project (DSDP), Ocean Drilling Program
(ODP), and Integrated Ocean Discovery Program (IODP) (e.g., Zachos et al., 1996; Pekar and Miller, 1996; Salamy and
Zachos, 1999; Coxall et al., 2005; Bordiga, et al., 2015; Hutchinson et al., 2021 and references therein). In contrast, changes
associated with the EOT in marginal and epicontinental seas have been the focus of relatively few studies (e.g., Pearson et al.,
2008; Ozsvárt et al., 2016; van Der Boon et al., 2019; Dickson et al., 2021). Nevertheless, geochemical and sedimentary data
from these shallow regions can offer valuable insights into the impacts of the EOT, including Antarctic glaciation and cooling,
within restricted marine environments influenced by local factors such as freshwater influx, salinity variations, weathering,
erosion and terrestrial (sediment and carbon) fluxes.

This study addresses the gap in understanding the EOT conditions in epicontinental seas by revisiting and reanalyzing an
Eocene-Oligocene Boundary (EOB) section in the Karaburun area of northern Türkiye (Figure 1a). The Karaburun section
with its exceptionally well-preserved and diverse assemblages of microfossils (e.g. Simmons et al., 2020; Sancay and Batı,
2020) provides an excellent opportunity to investigate the distribution of key latest Eocene - earliest Oligocene organic walled
dinoflagellate cyst (dinocyst) and calcareous nannofossil index species. Furthermore, it sheds light on how climatic changes
influenced stratigraphic sequences in the eastern Paratethys Sea—the largest Cenozoic epicontinental sea, with no modern
analogue. To investigate these processes, the EOB section at Karaburun is analyzed using marine palynology emphasizing
dinocysts and calcareous nannofossil biostratigraphy, as well as high-resolution stable oxygen and carbon isotope analyses of
benthic (*Cibicidoides* spp.) and planktic (*Turborotalia ampliapertura*) foraminifera. U-Pb dating of a tuff layer within the
section further constrained the age model, complementing the biostratigraphic and chemostratigraphic data, also allowing
sequence stratigraphic interpretations. The findings are compared with the EOT records from other Paratethys sites and global
oceans to discern the regional and global climatic and oceanographic effects.

## 74 2 Geological Setting

The interplay of paleoclimatic and tectonic processes fragmented the largely enclosed Paratethys water body into numerous
sub-basins, separated by narrow, shallow gateways and land bridges (Palcu et al. 2022). Extending from southern Germany to
China (Figure 1b), Paratethys encompassed three distinct regions. The Western and Central Paratethys are characterized by
active tectonics and comprised smaller, short-lived basins. The Western Paratethys included the western Alpine foreland basin,
while the Central Paratethys included sub-basins spanning from Austria to Romania (e.g., Popov et al., 2004). In contrast, the
Eastern Paratethys, centered around the Black Sea and Caspian Sea basins, evolved within a tectonically stable region (Popov
et al., 2004).

The Karaburun area is situated along the southern margin of the Western Black Sea Basin, a back-arc basin formed during the
late Cretaceous as a sub-basin of the Paratethys Sea (Okay and Nikishin, 2015) (Figure 1c). Since its formation, the Western
Black Sea Basin has experienced continuous subsidence, resulting in a sedimentary thickness exceeding 14 km (Okay et al.,
2019). To the south of the Karaburun area lies the Thrace Basin (Figure 1a), which is younger and characterized by Eocene-
Oligocene clastic fill deposits, reaching a maximum thickness of approximately 9 km in its central region (Turgut, 1991).
During the Eocene and Oligocene, the Strandja Massif—a polydeformed, deeply eroded orogenic belt composed of
metamorphic and magmatic rocks—formed a paleo-high that separated the Western Black Sea Basin from the Thrace Basin
to the south (Cattò et al., 2018) (Figure 1a). The only marine connection between these basins was through the Çatalca Gap
(Figure 1a), located west of İstanbul, where sedimentation abruptly ceased during the early Oligocene due to an uplift event
(Okay et al., 2019). This region represents the sole contact point between the Eocene-Oligocene sequences of the Black Sea
and Thrace Basins (Okay et al., 2019).

## 94 3 Regional Stratigraphy

The Soğucak Formation, characterized by shallow marine, massive reefal limestone, underlies the uppermost Eocene–
lowermost Oligocene hemipelagic deposits in both the Thrace Basin and the Karaburun area (Figure 1b). In the Karaburun
area, the Soğucak Formation has been dated to the Priabonian based on benthic foraminiferal biozonation (Yücel et al., 2020).
Overlying this formation is a 120-meter-thick sedimentary succession from the latest Eocene to early Oligocene, predominantly
composed of hemipelagic marls and carbonates (Figure 1c). This sequence also includes intermittent submarine fan deposits,
debris flows, slumps, pebbly sandstones, and conglomerates near the top. A prominent tuff layer within this succession is
linked to a significant Rupelian volcanic event originating from the Rhodope Massif (Marchev et al., 2024). Although the
hemipelagic succession in the Karaburun area has often been referred to as the Ceylan Formation—following the terminology
used in the Thrace Basin (e.g., Natal'in and Say, 2015)—we adopt the designation "İhsaniye Formation," as recommended by
Okay et al. (2019) and Simmons et al. (2020). While prior studies assigned an early Oligocene age to the İhsaniye Formation
in the Karaburun area (e.g., Less et al., 2011; Okay et al., 2019; Simmons et al., 2020), our findings refine its age to the latest
Eocene–early Oligocene. This revision is based on the calcareous nannofossil and dinocyst biostratigraphy, stable oxygen and
carbon isotope analyses, and U-Pb dating of the tuff layer.

## 4 Material & Methods

### 4.1 Lithostratigraphy

Eocene and Oligocene deposits are well-exposed in 50-meter-high cliffs along the Black Sea coast in the Karaburun area
(Figure 1d). These deposits have been the focus of recent studies (e.g., Okay et al., 2019; Sancay and Bati, 2020; Simmons et
al., 2020; Tulan et al., 2020), which documented their paleoenvironment, biostratigraphy, and source rock potential. To build
upon these studies, we revisited the area and measured three adjacent stratigraphic sections—designated as KR1, KR2, and
KR3 (Figure 1d). By integrating these sections, we constructed a composite Karaburun section comprising hemipelagic
deposits of the İhsaniye Formation (Figure 2).

The studied sediments predominantly consist of hemipelagic light gray marls, dark brownish clays, thin- to medium-bedded
light gray, whitish, and beige carbonates, calcareous siltstones, and sandstones, which occasionally display planar lamination.
The sequence also includes submarine fan (turbiditic) conglomerates, as well as debris flow and slump deposits toward the
top. The hemipelagic fine-grained deposits contain rich microfossil assemblages of planktic and benthic foraminifera,
calcareous nannofossils, and dinocysts, indicating a latest Eocene–early Oligocene age (this study). The submarine fan deposits
are characterized by thin to medium thick, erosive-based conglomeratic beds with mainly carbonate pebbles, organic matter,
and microfossil fragments (e.g., foraminifera and shell debris) and intercalated with thin silty layers. They often grade vertically
into sandstone layers. Although these submarine fan deposits exhibit variable lateral thicknesses, they provide solid key
horizons for correlation of the subsections (e.g. correlation of the KR1 and KR3 subsections). Brownish organic-rich clay
layers occasionally contain red-yellow nodules. Pyrite is commonly found in these organic-rich layers. Additionally, pyritized
coral fossils are rarely observed in these organic-rich clay deposits. A distinctive white tuff layer at approximately 71.5 m
within the composite section was sampled for U-Pb zircon dating (see section 4.5) (Figure 2). Debris flow horizons, with a
maximum thickness of 5 meters, exhibit channel geometries and primarily consist of carbonate pebbles. These horizons
increase in frequency toward the topmost 20 meters of the succession. This study focuses on the lower and middle portions of
the section including the EOT, ending around the tuff layer (Figure 2), and does not include analysis of the uppermost part of
the succession including the debris flow deposits.

## 4.2 Sequence Stratigraphy

We analyzed various surfaces that indicate either a seaward or landward shift of successive facies belts, including erosive
surfaces which could be equivalent to subaerial unconformities on land, transgressive surfaces, and maximum flooding
surfaces. These surfaces demarcate the boundaries of different systems tracts—lowstand, transgressive, and highstand—which
together form the depositional sequences (Catuneanu, 2006 and references therein).

In addition to the identification of the systems tracts based on the recognition of key surfaces, sedimentary facies and
microfossil assemblages have been utilized to reconstruct past water depths and identify sea-level evolution, typically indicated
by shifts towards offshore (or onshore) characteristics. The distribution and relative abundance of planktic and benthic
foraminifera have further been employed to discern variations in relative sea level. Additionally, the relative abundance of
lagoonal and inner neritic dinocysts, combined with the distribution patterns of associated brackish and terrestrial
palynomorphs as well as the grain size of the deposited sediments, were examined to assess proximity to the coast.

## 4.3 Biostratigraphy

### 4.3.1 Calcareous nannofossils

The study on calcareous nannofossil assemblages was carried out on 84 samples, prepared at the Department of Earth Sciences
of the University of Milan (Italy), following the smear-slide standard technique described by Bown & Young (1998).
Calcareous nannofossils were analyzed using an Axioscop Zeiss light microscope (LM) at 1250X magnification. Preservation
of the specimens was generally good, as indicated by the presence of holococcoliths, coccospheres and small coccoliths.
Quantitative analysis was performed by counting 300 specimens per sample, in a variable number of fields of view, depending
on the nannofossils total abundance. Nannofossil frequency data were converted into the number of specimens per square
millimeter for the evaluation of the biostratigraphic signal, and into percentages to estimate the paleoecological significance
of the assemblage variations. The position of biohorizons recognized in this study is based on abundance patterns of index
species, according to Agnini et al. (2014) and Viganò et al. (2023a) and are labelled as follows: Top (T): the highest occurrence
of a taxon, Base common and continuous (Bc) and Top common and continuous (Tc): the lowest and highest common and
continuous occurrence of a taxon. For calcareous nannofossils taxonomy, we refer to Perch-Nielsen (1985), Agnini et al.
(2014), and the Nannotax web library (https://www.mikrotax.org/Nannotax3). The biostratigraphic schemes adopted here are
those of Martini (1971) and Agnini et al. (2014).

### 4.3.2 Marine Palynology - dinocysts

For marine palynological analysis, emphasizing dinocysts 42 samples were prepared at Petrostrat laboratories (Conwy, Wales, UK; sections KR1 and KR2), and another 10 (from the sub-section KR3) at Utrecht University laboratories, according to typical palynological processing techniques (see e.g., Cramwinckel et al., 2020). This involves freeze-drying and precision weighing, subsequent HCl and HF treatments, followed by sieving residues over a 15 μm mesh sieve, before slides were produced for light microscopy from the residues. Samples are spiked with a known amount of *Lycopodium clavatum* spores to allow for absolute quantitative analysis (Stockmarr, 1972). After a broad palynofacies characterization (non-quantitative), light microscopical analysis included counting of broad categories of aquatic and terrestrial palynomorphs up to a minimum of 100 identifiable dinocysts per sample. Fragments of palynomorphs identifiable or not (viz, fragments of indeterminable palynomorphs to e.g., fragments of – therefore - indeterminable dinocysts, and including fragments of inner linings of benthic foraminifera), were quantified as well (see Table S1).

For dinocyst taxonomy, we refer to that cited in Williams et al. (2019), except for taxa belonging to the Wetzelielloideae (see discussion in Bijl et al., 2017). All materials are stored in the collection of the Marine Paleoceanography and Palynology group, at the Laboratory of Palaeobotany and Palynology (Utrecht University, Faculty of Geosciences).

## 4.4 Geochronology

### 4.4.1 U-Pb dating

A 30 cm-thick volcanic tuff layer was identified at the 71.5 m level of the KR composite section, serving as a key marker for constraining the age of the deposits. Three kilograms of tuff material were crushed and zircon crystals were separated by standard heavy liquid techniques and mounted in epoxy resin. Thirty five zircon crystals were dated via U-Pb at the Envitop analytical facility at CEREGE using an Element XR ICP-MS connected to a NWR193 laser (ArF 193 nm) ablation system. Zircon crystals were ablated with a 25-micron spot diameter, a 15 Hz pulse repetition rate, an energy fluence of 1.5 J/cm², and a carrier gas flow of 0.975 L/min. Data reduction, date, and date uncertainty calculations were conducted with an in-house MATLAB script. We applied rigorous filtering based on zircon morphology, U-Pb concordance, and common Pb content to ensure the reliability of the final age determination. The dataset was filtered for concordant grains using the Concordia distance method of Vermeesch (2021), which applies isometric logratios with a discordance threshold of 10 (SI units) and a reverse discordance threshold of 5 (SI units). Details about our U-Pb dating workflow and data reduction steps are given in Licht et al. (2024). The three zircon validation reference materials used during these sessions yielded offsets around TIMS ages < 1% in most cases, and < 2% otherwise. Out of the 35 analyzed zircon crystals, 12 yield concordant U-Pb ages (see Supplementary Table S3). The final Concordia age was calculated with concordant ages only using IsoPlotR (Vermeesch, 2018).

**4.5 Geochemistry**

**4.5.1 δ¹⁸O & δ¹³C analyses**

Measurements of stable oxygen (δ¹⁸O) and carbon (δ¹³C) isotopes on benthic foraminiferal (*Cibicidoides* spp.) and planktic foraminifera (*Turborotalia ampliapertura*) test fragments were performed at GEOMAR, Kiel on a Thermo Scientific MAT 253 mass spectrometer with an automated Kiel IV carbonate preparation device. The isotope values were calibrated versus the NBS 19 (National Bureau of Standards) carbonate standard and the in-house carbonate standard ("Standard Bremen", Solnhofen limestone). Isotope values in delta-notation (δ) are reported in ‰ relative to the VPDB (Vienna Peedee Belemnite) scale. The long-term analytical precision is 0.06 ‰ for δ¹⁸O and 0.05 ‰ for δ¹³C (1-sigma value). Replicate measurements were not done due to the low numbers of specimens found.

**5 Results**

**5.1 Biostratigraphy**

**5.1.1 Calcareous nannofossils**

So far a number of high-resolution studies on calcareous nannofossils during the EOT have been published, focusing on specific regions, including high latitudes (Villa et al., 2014) and mid- to low-latitudes (Bordiga et al., 2015; Fioroni et al., 2015; Villa et al., 2021; Jones et al., 2019). Recently, new studies on IODP sediments have further enhanced our understanding of this critical interval in the Paleogene paleoclimate history (Raffi et al., 2024, Viganò et al., 2023a, b, 2024a). Our study contributes to the knowledge of nannofossil biostratigraphy in this interval, providing detailed documentation of the EOT under marine conditions within the eastern Parathethyan realm (see Table S2 and Figure S9).

The extinction of *D. saipanensis* defines the base of Zone NP21 (Martini, 1971), which corresponds to the base of Zone CNE21 as described by Agnini et al. (2014). This species, as well as the "rosette shaped" *Discoaster*, is absent in the lowermost samples analyzed, indicating that its last occurrence (at 34.4 Ma) predates the studied interval.

The Base of common (Bc) *Clausicoccus subdistichus* group defines the onset of Zone CNO1 of Agnini et al. (2014), which corresponds to the upper part of Zone NP21 (Martini, 1971). The increase in abundance of this informal taxonomic group represents the most reliable nannofossil bioevent to approximate the EOB across different regions (*e.g.* Marino and Flores, 2002; Toffanin et al., 2013; Fioroni et al. 2015, Viganò et al., 2023a). In the studied area, this event is well-delineated, showing an abrupt increase in abundance from approximately 17 to over 100 specimens/mm² in sample N18, ca. 13 m from the base of the studied section (Figure 2).

The top (T) of *Ericsonia formosa* marks the base of Zone NP22 of Martini (1971) and the base of Zone CNO2 (Agnini et al., 2014). However, in the studied composite section, the precise position of this bioevent remains uncertain due to the rarity and

scattered occurrence of the taxon in its final range, nevertheless it is likely located before the end of the *C. subdistichus* acme (Figure 2). This Top of common (Tc) and continuous occurrence of the *C. subdistichus* group, was positioned before the top (T) of *E. formosa* by Agnini et al. (2014) in the biozonation adopted here. However, recent studies challenge this interpretation (Viganò et al., 2023a, 2024a). In fact, recent findings suggest that the top common and continuous (Tc) of this group occurs above the top (T) of *E. formosa*, as previously reported by Backman (1987) and Catanzariti et al. (1997). The top common and continuous (Tc) of *C. subdistichus gr.*, occurring early in Subchron C12r, has also been documented above the top (T) of *E. formosa* in the Pacific (Toffanin et al., 2013; Viganò et al., 2023b), the Atlantic (Bordiga et al., 2015, Viganò 2023b) and the Indian Ocean (Fioroni et al., 2015; Villa et al., 2021, Viganò 2023a). These studies indicate that *C. subdistichus* persists and remains common even after the T of *E. formosa*. In our dataset, the abundance of *C. subdistichus* exhibits a marked decrease (Tc) in the upper 5 meters of the studied section (Figure 2). Consequently, the T of *E. formosa* (i.e. the boundary between NP21 and NP22) should be positioned at some point prior to this bioevent. This interpretation is supported by data from dinocysts (section 5.1.2), which provide a better and more precise constraint for the stratigraphic position of the upper part of the investigated section.

## 5.1.2 Dinocysts

Our work builds on the recent integrated study by Simmons et al. (2020) and notably that by Sancay and Bati (2020), targeting the Karaburun area and outcrops using palynological approaches, emphasizing dinocysts. Their pioneering effort, using more locations and sections, but with much lower sample resolution, now show the need for a higher-resolution approach, particularly while considering the potential recognition of a continuous EOT interval. Therefore, here, we focus on the lower parts of the section, with our higher-resolution sampling.

Unfortunately, palynomorph preservation and fragmentation varies significantly across the section, ranging from (most often) very poor to only occasionally reasonable, and always typically heavily fragmented (Table S1; note the high number of undeterminable - fragments of - specimens).

The dominant palynological groups throughout the succession are organic linings of benthic foraminifera, and notably their fragments, besides dinocysts, and pollen and spores of terrestrial higher plants (Table S1). We also recovered several other aquatic algal taxa and acritarchs. These include representatives of e.g., fresh to brackish water elements like *Cyclopsiella*, *Pterospermella* and, *Tasmanites* spp., besides very small (< ~20 µm) psilate and skolochorate cysts (viz, 'acritarchs', and other 'small skolochorate cysts') of unknown ecology. Fungal spores and fruitbodies, as well as scolecodonts are occasionally encountered as well. In terms of palynofacies (palynodebris) composition and trends, the samples are all very similar in displaying a rich mix in mainly terrestrial plant-derived elements of varying sizes, combined with various amorphous materials. Truly opaque material is conspicuously absent. No trends or breaks are apparent from this visual, non-quantitative assessment (Table S1). Overall, these results match the findings by Sancay and Bati (2020).


Although the dinocyst assemblages are difficult to quantify because of preservation and fragmentation issues, taken together,
assemblages are highly diversified throughout, and essentially composed of well-known late Eocene to early Oligocene taxa
(See Figure S4, S5, S6, S7 and S8). The assemblages are quite comparable to those known from other EOT sections in the
larger Tethyan region, e.g., from central and northeast Italy (Brinkhuis and Biffi, 1993; Brinkhuis, 1994; Van Mourik and
Brinkhuis, 2005; Houben et al., 2012; Iakovleva, 2025), to North Africa (Egypt, El Beialy et al., 2019, Tunisia, Toricelli and
Biffi, 2001, and Morocco, Chekar et al., 2018, Mahboub et al., 2019; Slimani et al., 2019; Slimani and Chekar, 2023), and
further to the East, e.g., the Caspian Sea region (Bati, 2015), and Armenia (Iakovleva et al., 2024). In terms of robust dinocyst-
based age-assessment, best calibrated information is available from central and northeast Italy (e.g., Brinkhuis and Biffi, 1993
and follow-up studies), including a detailed zonal scheme for the EOT matched with magneto- and calcareous microfossil
stratigraphies. Based on the first, and last regional occurrence of *Glaphyrocysta semitecta* the base of the Gse and the Rac/Cin
zonal boundary of Brinkhuis and Biffi (1993) can be recognized in the Karaburun sections (Figure 2, Table S1). Furthermore,
based on the last occurrence of *Hemiplacophora semilunifera*, the Gse/Adi zonal boundary can be recognized as well (Figure
2). Remarkably, the important index species *Areosphaeridium diktyoplokum* is so far not recorded in any of the sub-sections
with confidence, hampering the recognition of the Adi/Rac zonal boundary. This is noteworthy, as elsewhere in the region, the
species is typically quite abundantly present in the deposits assigned to the absolute earliest Oligocene (as defined by the
extinction of the Hantkeninids, planktonic foraminifera; see e.g., Brinkhuis and Visscher, 1995; Van Mourik and Brinkhuis,
2005; Houben et al., 2012). Yet, the recognition of the other zonal boundaries allows confident correlation to the EOT interval,
throughout matching the assignments by calcareous nannofossils, and the typical benthic foraminifer $\delta^{18}$O-EOT-profile
(including the Oi-1a event – cooling during the early part of Subchron C12r) discussed further below.

These correlations are here bolstered by the spot-occurrence of *Svalbardella cooksoniae* in sample Y41 at 71 m, assigned to
the Cin Zone (Figure 2, Figure S5a, b, and c). This event was previously described from the central Italian EOT section, within
the same Cin Zone (Brinkhuis and Biffi, 1993). At high northern latitudes, this species ranges from the late Eocene way into
the Oligocene (e.g., Eldrett et al., 2004; Eldrett and Harding, 2009). Subsequent work noted that colder episodes during the
Oligocene likely induced equatorward migration of such typical high-latitude taxa (e.g., van Simaeys et al., 2005). In effect,
our finding reflects the earliest one of such migration pulses, an event well documented by Śliwińska and Heilmann-Clausen
(2011). These authors showed that *Svalbardella cooksoniae* is consistently present in the same narrow interval calibrated to
the basal Subchron C12r, close to the NP21/NP22 boundary, in many high- and mid-latitude Northern Hemisphere sections,
ranging from the Greenland Sea in the north to Italy in the south. Moreover, they correlated this event to the Oi-1a oxygen
isotope maximum of Pekar and Miller (1996) and Pekar et al. (2002). Another interesting finding is specimens of the acritarch
*Ascostomocystis potane* in samples from the sub-section KR2. Documented from the basal Rupelian type section in Belgium
(Stover and Hardenbol, 1993), this further confirms assignment to the basal Oligocene.

## 5.2 U-Pb dating of the tuff

Concordia plot of the dated tuff is available in Supplementary Figure S10. The tuff sample yields an early Rupelian 32.55 ± 0.38 Ma (2σ) age, based on 11 out of the 12 dated zircons. The age of the tuff layer aligns with the biostratigraphic dating, which places the top of the KR composite section within the Cin dinoflagellate cysts zone (Figure 2).

## 5.3 Sequence Stratigraphy

Based on lithological, sedimentological, and paleontological characteristics, we interpret the depositional environment as a shelf setting exhibiting littoral and neritic characteristics, depending on fluctuating sea levels. Our sequence stratigraphic analysis identifies ten distinct depositional sequences within this setting (Figure 3). The lower part of the section, up to approximately 11 m, comprises three depositional sequences (S1, S2, and S3), characterized by intercalated pebbly/conglomeratic layers, marls, and claystones (Figure 3). The Lowstand Systems Tracts (LSTs) and Highstand Systems Tracts (HSTs) contain both pebbly/conglomeratic layers and fine-grained deposits, whereas the Transgressive Systems Tracts (TSTs) are represented exclusively by fine-grained marls and claystones. The relatively low abundance of lagoonal dinoflagellate cysts (20–40%) in this interval suggests a distal position far from the coastline. Between approximately 20 m and 23 m, within Sequence 5, the depositional setting represents the deepest marine conditions, marking a more distal position relative to the coastline. This interval corresponds to the TST within Sequence 5.

At around 23 m, just below the last conglomeratic layer, a maximum flooding surface marks the base of the HST within Sequence 5, coinciding with the highest percentage of lagoonal dinoflagellate cysts. This increase in lagoonal dinoflagellates (up to 60–70%) continues into the LSTs of Sequences 6 and 7, indicating a more proximal position near the coastline during the Eocene-Oligocene Glacial Maximum (EOGM).

The upper part of the section, including the Early Oligocene Cooling (see Section 6.3), consists of four sequences (S7, S8, S9, and S10). A notable difference in the thickness of the depositional sequences is observed between the lower section (up to ~11 m) and the overlying sequences (from ~11 m to the top) (Figure 3). This variation in depositional thickness could be related to obliquity forcing during the latest Eocene, a process that has been identified in several terrestrial and marine records (e.g., Abels et al., 2011; Jovane et al., 2006; Boulila et al., 2021).

Finally, we compared our reconstructed relative sea-level variations, based on the depositional sequences and systems tracts described above, with the global sea-level reconstruction of Miller et al. (2020) (Figure 3). In the Eastern Paratethys region, relative sea-level evolution during the latest Eocene and early Oligocene appear to follow a pattern parallel to global sea-level changes. (Note that the base of the Karaburun composite section is younger than 34.4 Ma, see Section 5.1.1.) These high-resolution sea-level evolution offer a refined reconstruction of Eastern Paratethys sea-level changes (e.g., Popov et al., 2010) and provide a valuable framework for future studies to further reveal the history of Paratethys sea-level evolution..

### 5.4 Geochemistry

### 5.4.1 δ¹⁸O and δ¹³C isotope analyses

The $\delta^{18}O$ values of benthic foraminifera (*Cibicidoides* spp.) in the Karaburun composite section range from -9.5‰ to 1.0‰, displaying distinct temporal variations throughout the sequence. At the base of the section, at 11.40 m, just before the EOB, a small positive peak (1.0‰) is noticeable (Figure 4). Following this, a pronounced and abrupt negative shift from -0.1‰ to -9.5‰ is evident just after the Eocene-Oligocene Boundary (EOB). Following the abrupt negative shift, the $\delta^{18}O$ values increase gradually and then sharply, reaching a peak of 1.0‰ at 19.60 m. This represents a two-phase increase in $\delta^{18}O$ during the EOT as in other EOT records (e.g., Katz et al., 2008), however, with a negative shift in between. Following the second increase, a negative shift to -2.3‰ occurs at 22 m, followed by a renewed increase in $\delta^{18}O$ values, forming a plateau that culminates at 0.6‰ by 36.75 m. The $\delta^{18}O$ values then decline, reaching a negative peak of -1.6‰ at 40.25 m, before gradually rising to 0.4‰ at 60 m, marking a second, shorter plateau. This plateau is interrupted by a decrease to -1.6‰ at 65.75 m, followed by a slight recovery to 0.5‰ at 69 m. In the uppermost portion of the section, $\delta^{18}O$ values drop to -2.4‰ at 74.50 m and then exhibit a modest increase toward the top of the section (Figure 4).

The $\delta^{13}C$ values of benthic foraminifera (*Cibicidoides* spp.) in the Karaburun composite section range from -0.7‰ to 2.2‰, exhibiting greater variability compared to the $\delta^{18}O$ values. However, similar to the $\delta^{18}O$ values, the $\delta^{13}C$ values also display a prominent shift towards more depleted levels (from 2.1‰ to 0‰) just after the EOB (Figure 5). The two-phase increase in $\delta^{13}C$ (from the EOB up to ca 24 m) is again evident, interrupted by a sharp decline (ca. 20 m) that corresponds precisely with changes in $\delta^{18}O$ (Figure 4 & 5). The $\delta^{13}C$ appears to lag behind by several tens of thousands years compared to the two-phase increase in $\delta^{18}O$ (e.g., Coxall et al., 2011). The increase just before the EOB between 11.20 m and 12 m represents a 1.3‰ shift, followed by a 1.5‰ increase between 13.20 m and 15 m and a 1.0‰ increase between 19.60 m and 23 m. Before the EOB, two more major positive carbon isotope excursions are observed at the base of the section: one between 1 and 7.5 m (1.0‰) and the second between 7.5 m and 11.20 m (0.1‰). All excursions, except the second at the base (between 7.5 m and 11.20 m), exhibit a significant positive shift of ≥1.0‰. After the excursion between 19.60 m and 23 m, the $\delta^{13}C$ values drop sharply (around 23–27 m) before exhibiting another series of positive carbon isotope excursions in the middle and upper parts of the section (Figure 5). Like the excursions at the base of the section, the excursions in the middle and upper parts also display positive shifts of approximately 1‰, with the most pronounced reaching 1.5‰. This pattern reflects a dynamic carbon cycle with notable variations throughout the sequence.

Due to the intermittent presence of planktic foraminifera *Turborotalia ampliapertura* along the composite section, the $\delta^{18}O$ and $\delta^{13}C$ records of planktic foraminifera exhibit some gaps (Figures 6 and S1). Nevertheless, the overall trends and shifts remain discernible. Similar to the $\delta^{18}O$ record of benthic foraminifera, the $\delta^{18}O$ of planktic foraminifera exhibits a clear positive shift at the base of the section at 11.40 m just before the EOB, although slightly smaller (1.2‰). Following a slight positive

shift (0.4‰) just after the EOB at 14.45 m, the most prominent and significant positive shift recorded by the benthic foraminifera is not fully captured in the planktic foraminifera record due to the absence of *T. ampliapertura* in the 4 m interval between 15.90 m and 19.90 m. However, the positive 2.3‰ shift observed from 15.90 to 19,90 m represents partly the major positive shift, albeit smaller than the shift recorded in the benthic foraminifera δ¹⁸O. An interval of ca. 7 m without any *T. ampliapertura* follows the pronounced positive shift between 19.90 m and 26.5 m, obscuring the δ¹⁸O record for that interval. Following the barren interval, a gradual increase of 0.9‰ is clearly observed, extending up to 36.75 m. In the interval between 36.75 m and 61 m, the δ¹⁸O record of planktic foraminifera fluctuates, exhibiting two significant alternating trends of decrease and increase. Following this fluctuation, a gradual decrease is observed at the top of the section, followed by a sharp decline.

A distinct decline in the δ¹³C record of planktic foraminifera (1.5‰) is observed just before the EOB at 11.60 m (Figures 6 and S2). Following this decline, δ¹³C values increase at the EOB, reaching to 1.5‰ at 15.90 m with a shift of 1.8‰. Due to the infrequent presence of *T. ampliapertura*, the interval between 15.90 m and 26.5 m is not fully represented; however, this interval includes the highest δ¹³C values observed. After these peak values, a gradual decrease is noted towards the top of the section, with six positive shifts interrupting this trend: between 34.50 m and 36.75 m, 45.20 m and 47 m, 49 m and 51 m, 56.50 m and 61 m, 62.25 m and 65.75 m, and 65.75 m and 70.10 m. At the top of the section, a sharp decline in δ¹³C values is noticeable.

## 5.5 Paleoenvironment & Paleoecology

### 5.5.1 Calcareous nannofossils

Calcareous nannoplankton are highly sensitive to environmental changes in their surface water habitats, and fluctuations in nannofossil assemblages are interpreted as responses to shifts in sea surface temperature (SST), nutrient concentrations, salinity, and other environmental factors (*e.g.*, Aubry, 1992; Winter et al., 1994) thereby reflecting palaeoceanographic perturbations. Numerous studies have explored the ecological tolerance of extinct taxa, establishing paleoecological preferences through biogeographic studies (*e.g.*, Wei and Wise, 1990) and comparison with diverse environmental proxies (*e.g.*, Villa et al., 2014). We discuss the behavior of several taxa within the Karaburun assemblage, based on paleoecological affinities outlined in previous works.

*Reticulofenestra daviesii* and *Chiasmolithus* spp. are considered cool-water taxa (Wei et al., 1992; Villa et al., 2008) with preference for eutrophic conditions (Villa et al., 2014, Viganò et al., 2024b). This paleoecologic group is recorded with very low relative abundances, with few positive peaks in the middle part of the studied section. *Cyclicargolithus floridanus,* a typical eutrophic open-ocean species (Auer et al., 2014), occurs with abundances reaching up to 50% in the lower and upper part of the studied section, suggesting high productivity conditions (Aubry, 1992, Dunkley Jones et al., 2008, Villa et al., 2021) (Figure S3). Small reticulofenestrids constitute a significant component of the assemblages, reaching over 60% in the middle part of

the section. They have been reported as dominant components of the nannoflora along continental margins (Haq, 1980). These
settings are typically characterized by eutrophic conditions, driven by continental runoff and/or riverine input. Consequently,
these small coccoliths are regarded as opportunistic taxa with broad ecological tolerance, yet particularly well-adapted to
nutrient-rich environments (Aubry, 1992) and indicative of increased availability of terrigenous nutrient (Wade and Bown,

388 2006).


The genus *Helicosphaera* has been linked to increased nutrient availability (De Kaenel and Villa, 1996; Ziveri et al., 2004).
Studies on extant coccolithophorids confirm the relationship of helicosphaerids with high primary productivity rates (Haidar
and Thierstein, 2001; Toledo et al., 2007), and their preference for near-shore environments (Ziveri et al., 2004; Guerreiro et
al., 2005). At the Karaburun section, this genus is recorded at low abundances but occurs consistently throughout the section
(Figure S3).

Evidence of nutrient availability is further supported by the presence of braarudosphaerids, which are associated with coastal,
low-salinity waters (Peleo-Alampay et al., 1999;  Thierstein et al., 2004; Konno et al., 2007), eutrophic conditions (Cunha and
Shimabukuro, 1997, Bartol et al., 2008) and the influx of terrigenous material (Švábenická, 1999). Braarudosphaerids are
rarely found in the open ocean and thrive under unusual marine conditions, demonstrating a tolerance for environmentally
stressed settings. Similar conditions are indicated by the presence of *Micrantholitus*, a taxon tipically associated with shallow
marine environments (Bown, 2005), reduced salinity, and eutrophic conditions (Street and Bown, 2005; Bown and Pearson
2009). These penthaliths occur from the base of the investigated section, albeit at low percentages and with a discontinuous
distribution, further suggesting eutrophication and reduced salinity (Figure S3).

The presence of Ascidian spicules, with their highest and continuous  occurrence in the middle-upper part of the section, also
points to a shallow-marine depositional setting (e.g. Varol, 2006; Ferreira et al., 2019)  and high surface-water productivity
(Toledo et al., 2007). Furthermore, the relatively common occurrence of holococcoliths (mainly *Lanternithus minutus* and
*Zigrablithus bijugatus*), *Pontosphaera* spp. and *Helicosphaera* spp., taxa prone to dissolution (Bown, 2005; Monechi et al.,
2000) reinforces the interpretation of a shallow-water environment.
**5.5.2 Marine palynology - dinocysts**
For the analysis of the marine palynological assemblages, emphasizing dinocysts, we rely on the taxonomical and ecological
dinocyst groups derived from modern distributions (e.g., Zonneveld et al., 2013; Marret and De Vernal 2024) and empirically
based paleoecological information or the Paleogene dinocysts following previous works (e.g., Brinkhuis, 1994; Pross and
Brinkhuis, 2005; Sluijs et al., 2005; Frieling and Sluijs, 2018). However, as mentioned above, the assemblages are generally
too poorly preserved to allow for detailed quantitative considerations. Yet, given the overall quantitative characteristics of the
studied samples, viz, substantial terrestrial input, and consistent dominances of taxa empirically known from restricted marine

to inner neritic (incl. lagoonal) like the goniodomid-group of dinoflagellate cysts (in this case e.g., *Homotryblium*, *Polysphaeridium*, *Heteraulacacysta*, *Eocladopyxis* spp.), and the peridinioids (*Lentinia*, *Phthanoperidinium*, *Senegalinium*, and *Deflandrea* spp.), neritic to outer neritic (e.g., *Areoligera*, *Glaphyrocysta*, *Enneadocysta*, *Spiniferites* and *Operculodinium* spp.) combined with a small, but consistent contribution from offshore, oceanic taxa like *Impagidinium* and *Nematosphaeropsis* spp. points to an essentially open marine, offshore, hemipelagic setting, comparable to e.g., the central Italian sections (cf. Brinkhuis and Biffi, 1993).

Despite the issues with preservation throughout, a percentage-plot of fresh water tolerant, and restricted to inner neritic marine taxa vs more offshore taxa still reveals stronger influxes in the latter part of the Gse, and within the Adi Zone (Table S1; see above, and compare e.g., Frieling and Sluijs, 2018, and Sluijs and Brinkhuis, 2024). In effect, this aspect matches the records from elsewhere (e.g., the Italian sections), and was earlier interpreted to reflect general eustatic sea level lowering associated with the Oi-1 stable isotope event reflecting the earliest glaciation of Antarctica (e.g., Brinkhuis, 1994). In terms of temperature/climatic changes, the conspicuous increase in Gymnospermous (conifer) bisaccate pollen input may be significant as well. Again, a similar trend was noted in the Italian sections across the EOT (Brinkhuis and Biffi, 1993; Brinkhuis, 1994).

## 6 Discussion

### 6.1 Age control overview

An initial age model for the KR composite section was constructed using tie points derived from nannofossil and dinocyst biozonations, combined with U-Pb dating of a tuff layer at 71.5 m (Figure 2). Additional age constraints were obtained by aligning the Karaburun benthic foraminifera $\delta^{18}O$ data with the high-resolution benthic foraminifera $\delta^{18}O$ record from the Atlantic sites 522 and 1263 (Figure 4). This alignment was achieved by identifying corresponding features in the isotope records, such as positive and negative shifts and their amplitudes. The carbon isotope record of benthic foraminifera provided independent validation of this tuning (Figure 5). The $\delta^{13}C$ benthic foraminifera data from the Karaburun area was correlated and aligned with global high-resolution benthic foraminifera $\delta^{13}C$ records from deep-sea sites, including 1218 (equatorial Pacific), 689 (sub-Antarctic Atlantic), 522 (South Atlantic), and 744 (southern Indian Ocean). Similarly, the planktic foraminifera provided a further confirmation for our age model (Figures S1, S2). The planktic foraminifera $\delta^{18}O$ and $\delta^{13}C$ data from the Karaburun area were correlated and aligned with high-resolution $\delta^{18}O$ and $\delta^{13}C$ records of planktic foraminifera in hemipelagic sediment cores retrieved from the African margin of the Indian Ocean (Tanzania Drilling Project sites 12 and 17, Pearson et al., 2008). All ages were assigned following the integrated magneto-biostratigraphic GTS2012 timescale. Our geochemical results indicate that the increases in $\delta^{18}O$ and $\delta^{13}C$ observed during the EOT at mid- and high-latitude sites in the South Atlantic, Southern Ocean and Pacific are also present in the Paratethys, verifying that these signals are genuinely global and valuable for stratigraphic correlation.

According to the constructed age model, the base of the section dates to the late Eocene (late Priabonian). The EOB is identified at 12.75 m. The middle and upper parts of the composite KR section correspond to the early Oligocene (early Rupelian) (Figure

2). The revised age constraints established in this study offer a robust chronostratigraphic framework for the latest Eocene to
early Oligocene interval of the Karaburun section, surpassing the accuracy of prior studies (e.g., Less et al., 2011; Okay et al.,
2019; Simmons et al., 2020). However, we have to note that the absence of the typically abundant index taxon *Areosphaeridium*
*diktyoplokum* in all studied sub-sections currently hampers precise recognition of the Adi/Rac zonal boundary and may reflect
local paleoenvironmental or taphonomic conditions, or alternatively, point to a slight stratigraphic gap or condensed interval,
a possibility that merits further investigation.

## 6.2 The EOT

At the deep Atlantic Site 522 and Pacific Site 1218, the late Eocene Event is marked by a transient interval of positive $\delta^{18}O$
values, reflecting a short-lived cooling or glacial episode (Hutchinson et al., 2021) (Figure 4). This isotopic shift measures
approximately 0.6‰ and 0.4‰ at Site 522 and Site 1218, respectively. Similarly, the base of the KR composite section exhibits
an increase of 0.7‰ in benthic foraminifera $\delta^{18}O$ values at approximately 5.5 m, which we interpret as evidence of the Late
Eocene Event (Figure 4). The onset of this event coincides with the extinction of *Discoaster saipanensis* at 34.44 Ma at Site
1218. Based on calcareous nannofossil data (i.e., the absence of *D. saipanensis*), the base of the KR section is inferred to be
younger than 34.44 Ma, supporting this correlation. The Late Eocene Event represented by this 0.7‰ positive shift in $\delta^{18}O$
values at approximately 5.5 m marks the onset of the EOT in the KR composite section (e.g., Hutchinson et al., 2020) (Figure

464  4).


The initial $\delta^{18}O$ step increase, occurring just before the EOB, has been identified as Step 1 in some records (e.g., EOT-1 in
Katz et al., 2008; Precursor Glaciation in Scher et al., 2011). The first 1.0‰ $\delta^{18}O$ increase observed in the KR composite
section at 11.40 m is interpreted as Step 1 as in the previous records (e.g., EOT-1 in Katz et al., 2008; Precursor Glaciation in
Scher et al., 2011) (Figure 4). A similar $\delta^{18}O$ increase of 0.9‰ is recorded in the Alabama St. Stephens Quarry core (Miller et
al., 2008). The onset of Step 1 is dated to 34.15 Ma, with an estimated duration of approximately 40 kyr (Hutchinson et al.,

471  2020).


The Earliest Oligocene Oxygen Isotope Step (EOIS) represents a rapid $\delta^{18}O$ increase (0.7‰ or more) occurring well after the
EOB, within the lower part of Chron C13n (Hutchinson et al., 2020). The peak $\delta^{18}O$ is recorded at approximately 33.65 Ma,
with the entire EOIS lasting around 40 kyr. In the KR composite section, the positive shift associated with EOIS is
approximately 2‰, peaking at 1‰ at 19.60 m, marking the end of the EOT (Hutchinson et al., 2020) (Figure 4).

The Early Oligocene Glacial Maximum (EOGM) is characterized as a prolonged period of cold climate and glaciation during
the early Oligocene, corresponding to the most of paleomagnetic Subchron C13n (Hutchinson et al., 2020) (Figure 4). It spans
from approximately 33.65 Ma to 33.16 Ma, lasting about 490 kyr. Correlation between the Karaburun data and global deep-
sea records was achieved by aligning the peak-to-peak δ¹⁸O stratigraphic intervals, starting at the top of the EOIS at 19.60 m
and extending to another peak at 36.75 m (0.6‰) corresponding to the top of Subchron C13n (Figure 4).

Overall, the δ¹⁸O record of benthic foraminifera from the Karaburun composite section closely mirrors global δ¹⁸O trends from
deep-sea sites, except for a sharp decrease observed just after the EOB (Figure 4). The EOT signal is clearly recorded in the
δ¹⁸O benthic foraminifera data from the Karaburun composite section. However, the relatively lower δ¹⁸O values and sharp
decrease just after the EOB are attributed to regional conditions in the Paratethys Sea, as discussed in Section 6.4.
**6.3 The early Oligocene cooling**
The presence of the cold-water dinoflagellate *Svalbardella cooksoniae* within a brief interval of the early Oligocene in the
North Atlantic and Western Neo-Tethyan realms has been previously documented and linked to the Oi-1a oxygen isotope
maximum (Śliwińska and Heilmann-Clausen, 2011). This oxygen isotope maximum representing a cooling event occurs during
the early part of Subchron C12r, near the NP21/NP22 boundary. In the KR composite section, the *Svalbardella cooksoniae*-
bearing sample aligns with an oxygen isotope maximum at approximately 71 m, occurring during the early phase of Subchron
C12r (Figure 4). Consequently, the boundary between NP21 and NP22 is likely located near this level. In the North Sea, the
*S. cooksoniae* event was identified at the top of the regressive systems tract (OSS-21 RST) in the 11/10-1 well (Śliwińska,
2019a). Similarly, in the KR composite section, the *S. cooksoniae* event is positioned at the top of a Lowstand Systems Tract,
in agreement with the North Sea data (Figure 3). Strontium isotope analyses by Jarsve et al. (2015) suggest an age of 32.66
Ma for this event. Our U-Pb dating of the tuff layer located just above the *Svalbardella* spp.-bearing interval yields an age of
32.55 ± 0.38 Ma, aligning closely with the strontium-based age reported by Jarsve et al. (2015). These findings further support
the interpretation of Śliwińska and Heilmann-Clausen (2011) that the earliest Rupelian *S. cooksoniae* interval across the
Tethys, Central Europe, the North Sea Basin, the Norwegian-Greenland Sea, and the Eastern Paratethys is coeval with the Oi-
1a event and corresponds to a significant sea-level fall (Figure 3).

In support of the geochemical evidence provided by δ¹⁸O values in benthic foraminifera, a notable increase in gymnospermous
(conifer) bisaccate pollen is clearly observed at the KR composite section during the EOT, EOGM, and Oi-1a events (Figure
3). This increase is likely associated with cooling and glaciation events occurring during these intervals, as previously
suggested by Brinkhuis and Biffi (1993) and Brinkhuis (1994).

The early Oligocene cooling event (Oi-1a) was previously dated to 32.8 Ma by Pekar et al. (2002). At the KR composite
section, the peak δ¹⁸O values in benthic foraminifera (~0.4‰) observed around 58–60 m are interpreted as representing the
Oi-1a event. Our age model corroborates the age proposed by Pekar et al. (2002), further supporting a timing of 32.8 Ma for
these peak values.

## 6.4 The regional & global effects in the Paratethys

The benthic foraminiferal oxygen and carbon isotope records from the Karaburun area closely resemble deep-sea records from Atlantic sites 522 and 1263 during the latest Eocene and early Oligocene (Figures 4, 5). However, a notable distinction is the pronounced negative $\delta^{18}O$ shift just after the EOB, a feature characteristic of the Paratethys region (Figures 4 and 7) which will be discussed in the following. Firstly, it is noticeable that the overall benthic and planktic foraminifera $\delta^{18}O$ values are more depleted than global records from the EOT. These depleted values likely reflect a regional effect rather than diagenetic alteration, as the exceptional preservation and glassy appearance of the foraminiferal shells from the KR composite section and other Paratethys sites (e.g., Ozsvárt et al., 2016) suggest minimal recrystallization. A major diagenetic overprint affecting the entire basin is also improbable given the differing tectonic and depositional histories across the Paratethys sub-basins. Additionally, the observed timescale (<100 ka) and the significant magnitude of changes in proxy records from the Paratethys Basin are unlikely to be explained by regional tectonic processes. Instead, factors such as basin restriction, enhanced precipitation and/or freshwater input due to increased runoff or changing hydrological conditions during the EOT seem more plausible explanations. Similar $\delta^{18}O$ depletion has been documented in other marginal basins during the EOT (e.g., Pearson et al., 2008; De Lira Mota et al., 2023), further supporting a localized effect in semi-restricted environments due to local hydrology and climate. Indeed, these values are consistent with the isotopic composition of meteoric waters at mid-latitude coastal regions ($\delta^{18}O \sim$ -5‰ to -10‰; Dansgaard, 1964; Gat, 1996). During the Rupelian (35–31 Ma), the dominant influence of Atlantic derived westerlies likely brought increased precipitation with a depleted $\delta^{18}O$ signature to the western and central Paratethys. This interpretation was supported by $\delta^{18}O_{PO_4}$ values from herbivore tooth enamel, which reflect the depleted isotopic composition of drinking water (Kocsis et al., 2014). The enhanced precipitation was likely due to the intensification of the westerlies and the reorganization of oceanic and atmospheric circulation (e.g., Hou et al., 2022). As global cooling progressed and the meridional temperature gradient steepened, the westerlies strengthened, enhancing vapor transport from the Atlantic into Eurasia, contributing to higher precipitation in the western and central Paratethys (e.g., Koscis et al., 2014; Li et al., 2018). Modeling studies further suggest prevailing westerly winds during winter at mid-high latitudes in the Rupelian (Li et al., 2018). The "continental effect," where $\delta^{18}O$ in meteoric water becomes progressively fractionated with increasing transport distance from the Atlantic, likely contributed to more negative $\delta^{18}O$ values in the Karaburun area (e.g., Kocsis et al., 2014). Basin geometry, particularly water depth, plays a crucial role in evaporation dynamics within a semi-isolated system like the Paratethys. The Western Paratethys, characterized by deeper basins, had a higher heat capacity and greater thermal inertia, leading to moderated surface temperatures and lower evaporation rates. In contrast, the shallower Eastern Paratethys had lower heat capacity, allowing for rapid surface warming, especially in warmer seasons, which enhanced evaporation. Additionally, wind-driven effects likely played a significant role. In deeper basins, wind-driven mixing redistributed heat within the water column, reducing extreme surface warming and stabilizing evaporation rates. In shallower areas, reduced mixing allowed surface waters to remain warmer, leading to increased evaporation, particularly under strong seasonal winds. Hence, further east, evaporation over the shallower Eastern Paratethys may have added moisture to westerly air trajectories,

resulting in relatively less negative δ¹⁸O values in the Northern Caucasus (Karaburun, Belaya and Chirkei sections in Figure
7) and increased inland precipitation (Figure 7). A similar precipitation gradient, with wetter conditions in the western-central
Paratethys and drier conditions in the east, is also evident in an Oligocene climate reconstruction based on plant macrofossil
data (Li et al., 2018). Additionally, the increased freshwater input in the Paratethys at the EOB could be plausibly explained
by the major sea level fall and falling base level, driven by glacio-eustasy associated with the growth of Antarctic ice sheets
during the EOT. The reorganization of rivers due to the falling base level would have introduced fresh water into the
depositional epicenters of the Paratethys. Combined, the effects of regional  hydrological change and the base level fall due
the global major sea level fall at the EOB might have resulted in the depleted δ¹⁸O values observed in the Paratethys. Despite
the localized variations in the depletion of δ¹⁸O values, the relatively consistent δ¹⁸O depletion observed across Paratethys
sections suggests a uniform basin-wide isotopic background. This consistency allows for reliable identification of major trends
and isotopic excursions in the Paratethys Basin during the EOT.

Second to notice is that before the EOB a parallel trend could be observed for the benthic and planktic foraminifera δ¹⁸O and
δ¹³C values (Figure 6). Particularly during the Step1 same trends in isotopic shifts could be clearly recognized. Most
significantly, just after the EOB a distinct contrasting trend between benthic and planktic foraminifera δ¹⁸O and δ¹³C could be
noticed (Figure 6). These contrasting trends between benthic and planktic foraminifera δ¹⁸O and δ¹³C just after the EOB suggest
significant stratification and a reduction in vertical mixing. The pronounced negative δ¹⁸O shift in benthic foraminifera likely
reflects a significant influx of isotopically light cold freshwater into the bottom waters. The slight increase in planktic
foraminifera δ¹⁸O at the same time suggest that the surface waters might have become relatively saline due to evaporation
exceeding freshwater input in the surface layer, which would increase δ¹⁸O values. Cold, freshwater inflow might have been
funneled into deeper areas of the basin, displacing or mixing with bottom waters. This might have been happened through the
submarine channels providing sediment-laden freshwater as underflows into the deep-marine turbiditic systems. However, this
would have required an anti-estuarine circulation model where marine saltwater flows upstream and overrides the freshwater
inflow. Indeed, an anti-estuarine circulation model for the early Oligocene Paratethys was proposed earlier (Dohmann, 1991)
which was later supported by Schulz et al. (2005) showing also increasing surface salinities due to evaporation of marine water
based on increasing di-/tri-MTTC ratios. An early anti-estuarine circulation model for the Paratethys aligns perfectly with our
abovementioned stable isotope data. The deep freshwater input could be explained by the early evolution of the Paratethys
(e.g., Schulz et al., 2005). Early evolution of the Paratethys was mainly controlled by narrowing seaways connecting it to the
Tethys Ocean which led to ingressions of cold boreal water from the north (through Polish straits), initially as undercurrents
(generating an anti-estuarine circulation) into the Eastern Paratethys first and then to Central and Western Paratethys (e. g.,
Schulz et al., 2005; Soták, 2010). In parallel to the δ¹⁸O values, a divergent trend could be observed between the benthic and
planktic δ¹³C values just after the EOB where benthic δ¹³C declines largely whereas the planktic δ¹³C shows an increase (Figure
6). In addition to the boreal fresh-water ingression, enhanced organic matter production due to increased nutrient input and
then the subsequent decomposition in the isolated, stratified Paratethys waters might have released light carbon into the bottom

waters. Increased $\delta^{13}C$ in planktic foraminifera could have resulted from elevated primary productivity driven by increased nutrient input, which preferentially removes isotopically light carbon from surface waters during photosynthesis, leaving the remaining carbon pool enriched in heavier carbon.

Initial boreal freshwater input by undercurrents then changes into ingressions of freshwater runoff as overflowing currents and diluting the former Paratethyan sea water (i.e., a change into estuarine circulation) (e.g., Schulz et al., 2005; Soták, 2010). This later ingression of overflowing freshwater runoff is likely due to an enhanced precipitation and represented as declining planktic foraminifera $\delta^{18}O$ and $\delta^{13}C$ values at ca. 15m (Figure 6). The $\delta^{18}O$ benthic foraminifera shows an increase during this time which is likely related to the cooling of the bottom waters where as $\delta^{13}C$ of benthic foraminifera shows a positive peak suggesting an enhanced organic carbon burial. The ingressions of freshwater runoff as overflowing currents likely formed a freshwater surface layer reducing ventilation of bottom waters. The formation of the freshwater surface layer and subsequent restricted mixing could have led pronounced stratification in the water column with isotopically lighter freshwater dominating the surface waters which is evidenced by more depleted $\delta^{18}O$ planktic foraminifera values (between 15 m and the top of the section) (Figure 6). The subsequent stratification in the water column would have exacerbated the buildup of oxygen-depleted conditions and the isotopically depleted carbon pool at the bottom. This would also favor sulfate reduction by microbial processes which produce further isotopically light carbon and reduce $\delta^{13}C$ values in an euxinic benthic environment. Consequently, stratification and reduced oxygenation must have enhanced the preservation of organic matter in bottom sediments.

The decrease in $\delta^{18}O$ of benthic foraminifera at ca. 22 m is possibly related to the slight warming recorded in the North Sea (Śliwińska et al., 2019b) in paleomagnetic Subchron C13n above the EOIS (Figure 6). The increase in $\delta^{18}O$ planktic foraminifera after ca. 25 m up to the ca. 40-41 m is likely related with further cooling during the EOGM. A sharp declining trend could be noticed for benthic foraminifera $\delta^{13}C$ during the onset of this interval (at ca. 25 m) suggesting a relatively less organic carbon burial. This was due to a decrease in primary productivity at the surface represented by lowering $\delta^{13}C$ of planktic foraminifera and lowered terrestrial input.

The relative sea-level in the Paratethys starts to lower after the ca. 40-41 m level (Figure 3) which is followed by another freshwater input likely due to enhanced precipitation at level ca. 45 m. This is evidenced by the depleted $\delta^{18}O$ planktic foraminifera values and an increase in terrestrial palynomorphs (Figures 3, 6). Once again this was followed by an increase in organic carbon burial represented by a peak in benthic $\delta^{13}C$ values at ca. 47 m.

During the Oi-1a cooling (between section levels ca. 45 m and 75 m) a long-term declining trend in benthic $\delta^{13}C$ is distinctive and suggests a decrease in organic carbon burial in the Karaburun area (Figure 6). Decreasing relative sea-level and related increase in bottom current velocities and wave action combined with a decrease in fresh water input should have likely

decreased the organic carbon burial suggested by decreasing benthic $\delta^{13}C$ at ca. 52 -53 m. Another freshwater ingression as
surface runoff could be seen at level ca. 56 m represented by a sharp decrease in planktic $\delta^{18}O$ and $\delta^{13}C$. It appears that the
global Oi-1a cooling signal dominates the upper part of the section between levels ca. 60 m and 75 m. At ca. 63 m a decrease
in planktic $\delta^{18}O$ and $\delta^{13}C$ values corresponds to surface freshwater input accompanied by increased organic carbon burial (peak
in benthic $\delta^{13}C$ values). A sharp decline in both benthic and planktic $\delta^{18}O$ and $\delta^{13}C$ values suggest a significant freshwater
influx  at level ca. 65 m. The uppermost peak in benthic $\delta^{13}C$ values at ca. 70 m represents and enhanced organic carbon burial
in the Karaburun area due to more favorable conditions for organic carbon sequestration provided by a relative sea level rise
(e.g., lower bottom current velocities and lower wave action).

The divergent trend between benthic and planktic $\delta^{13}C$ values indicates a highly stratified Paratethys Sea from time to time
and different surface and bottom carbon cycling processes after a change from anti-estuarine to estuarine circulation during
the EOT. These changes reflect both regional hydrological and basin reconfiguration (restriction) controls and global climatic
and eustatic shifts within the EOT. The global cooling during the EOT and Oi-1 must have amplified the stratification, reduced
ventilation and triggered local environmental shifts in the semi-enclosed Paratethys Basin. These environmental shifts provided
favorable conditions for the deposition of organic-rich fine-grained sediments with high Total Organic Carbon (TOC) values.

Overall, this contrasting pattern between the isotopic values of benthic and planktic foraminifera highlights the complex
interplay of global climate trends (Antarctic glaciation and global cooling) and regional factors (basin isolation and
hydrological changes) during the EOT. Our findings align with the proposed isolation of the Paratethys, driven by the
prolonged African-Arabian–Eurasian collision coupled with eustatic sea-level decline at the EOB and the cooling during the
EOT. In turn this led to the development of a distinct Paratethyan domain marked by mesophilic, humid climatic conditions
and intensified runoff (Popov et al., 2002). Moreover, the reconstructed relative sea-level evolution closely correspond to other
reconstructions of relative sea-level changes from the late Eocene to early Oligocene period (e.g., from the Neo-Tethys,
Brinkhuis, 1994;  North Sea, Jarvse et al., 2015), further reinforcing the presence of a global climatic signal, as well.
**6.5 Boreal water in the Paratethys during the early Oligocene and its paleoceanographic significance**
The prominent negative shift in $\delta^{18}O$ of benthic foraminifera likely representing the boreal water ingression recorded in the
Karaburun composite section shortly after the EOB (ca. 33.7 Ma) appears to be widespread across the entire Paratethys Basin
(Figure 7; Soták, 2010; Ozsvárt et al., 2016; Gavrilov et al., 2017; van der Boon et al., 2019). In the Eastern Paratethys, this
shift appears to have occurred abruptly. A similar negative shift in $\delta^{18}O$ of bulk carbonates is also recorded further west in the
West Alpine Foreland Basin (Chalufy section, Soutter et al., 2022), which connected the Western Paratethys to the
Mediterranean Tethys Ocean (Figures 7, 8). Hence, boreal water ingression into the Paratethys, beginning in the Eastern
Paratethys and progressively reaching the Central and Western regions, is clearly represented by this negative shift across the
entire Paratethys Basin just after the EOB. This process was likely linked to the restriction or closure of the Arctic-Atlantic

gateway and the onset of anti-estuarine circulation between the Atlantic and the Nordic Seas during the EOT. Proxy records indicate that the Eocene Arctic Ocean was significantly fresher than today, with salinities ranging from 20 to 25 psu and occasional drops below 10 psu (Brinkhuis et al., 2006; Kim et al., 2014; Waddell and Moore, 2008). The Arctic freshwater outflow into the North Atlantic may have inhibited deep-water formation during the Eocene (Baatsen et al., 2020; Hutchinson et al., 2018). Sea-level and paleo-shoreline reconstructions in the Nordic Seas support the hypothesis that the Arctic became isolated during the latest Eocene to early Oligocene due to the closure of the Arctic-Atlantic gateway (Hegewald and Jokat, 2013; O'Regan et al., 2011; Hutchinson et al., 2019). Recent evidence suggests that the deepening of the Greenland–Scotland Ridge (GSR) around the EOT (just before the EOIS, ca. 33.7 Ma) facilitated increased exchange between the Atlantic and the Nordic Seas, enabling the formation of anti-estuarine circulation and the salinization of North Atlantic surface waters (Abelson and Erez, 2017; Stärz et al., 2017). Consequently, the gradual restriction of Arctic-Atlantic connectivity, followed by the onset of anti-estuarine circulation driven by the deepening of the GSR, may have played a critical role in developing a robust AMOC (e.g., Coxall et al., 2018; Hutchinson et al., 2019). Together, the Atlantic-Arctic closure and onset of anti-estuarine circulation events could have triggered or intensified the Atlantic Meridional Overturning Circulation (AMOC) (e.g., Abelson and Erez, 2017; Coxall et al., 2018; Hutchinson et al., 2019). The onset of Nordic anti-estuarine circulation around the EOT might have likely influenced salinity gradients and circulation in connected basins like the North Sea and Paratethys, contributing to the freshening of the latter (Figure 8). Our data indicate the onset of anti-estuarine circulation in the Paratethys around 33.7 Ma, coinciding with the development of similar circulation patterns between the Nordic Seas and the North Atlantic (Abelson and Erez, 2017). We propose a hypothetical pathway for this circulation: warm surface water entering from the North Sea would have flowed into the eastern Nordic Seas, joining the warm surface waters from the North Atlantic. After losing heat, this water likely sank in the northern Nordic basin, flowed southward as deep water, and eventually reached the North Atlantic, the North Sea, the Paratethys, and the Mediterranean Tethys, forming an interhemispheric northern-sourced circulation cell (Figure 8a).

The onset of anti-estuarine circulation in the Paratethys was likely related to the deepening of the Greenland-Scotland Ridge (GSR), similar to the development of anti-estuarine circulation in the Nordic Seas. During most of the Paleogene, the GSR was shallower, restricting deep-water exchange between the North Atlantic and the Nordic Seas. A key feature of the modern AMOC is the formation of North Atlantic Deep Water, which flows over the GSR from the Nordic Seas. The initiation of Nordic anti-estuarine circulation events around the EOT likely enhanced deep-water formation, strengthening the Atlantic Meridional Overturning Circulation (AMOC) and establishing an interhemispheric northern-sourced circulation cell (e.g., Abelson and Erez, 2017; Coxall et al., 2018; Hutchinson et al., 2019). This suggests that by ~33.7 Ma, the Paratethys, with its anti-estuarine circulation, was integrated into this larger circulation system, contributing to global ocean circulation. Shortly after, subsequent geographic restrictions and hydrological changes during the EOT changed this anti-estuarine circulation to an estuarine circulation (Figure 8b). The closure of the Mediterranean Seaway—along with the narrowing of other seaways likely caused by the major sea-level fall during the EOIS and EOGM—and increased freshwater influxes from the continent

diluted the surface Tethyan waters, disrupted the anti-estuarine circulation pattern, and led to brackish surface salinities in the
Paratethys (e.g., Schulz et al., 2005; Soták, 2010).

Further evidence supporting boreal water ingression and circulation through the Nordic Seas, North Sea and Paratethys comes
from the distribution of *Svalbardella cooksoniae* in the Greenland, Norwegian, and North Seas, as well as in the Eastern
Paratethys during the Oi-1a cooling event (Śliwińska and Heilmann-Clausen, 2011). This suggests a possible migration
pathway for *Svalbardella* spp. The presence of *Svalbardella* in the Massicore and Monte Cagnero sections of central Italy
(Brinkhuis and Biffi, 1993; van Mourik and Brinkhuis, 2005) aligns with our interpretation of boreal water circulation
extending to the Mediterranean Tethys through the Paratethys (Figure 8). Sinking cold boreal freshwater likely propagated
through interconnected Nordic marine basins, reaching the Paratethys and eventually the Mediterranean Tethys (Figure 8).
The invasion of the Mediterranean Tethys by higher-latitude taxa around the EOB (Brinkhuis and Biffi, 1993) further supports
this circulation pathway. The incursion of boreal bottom waters into the Mediterranean Tethys during the EOT is corroborated
by a marked transient increase in deep-water ostracod *Krithe* and a decline in deep-water ostracod diversity in the Massignano
composite section (Slotnick and Schellenberg, 2013). The *Krithe* pulse and the subtle change in the deep-water ostracod fauna
reflects intensified thermohaline flow of cooler deep waters, likely linked to boreal freshwater circulation through the
Paratethys (e.g., Dall'Antonia et al., 2003; Slotnick and Schellenberg, 2013). Variations in seafloor ventilation and productivity
due to changes in paleoceanographic conditions of the Tethys during the late Eocene - early Oligocene described in previous
studies (Jovane et al., 2007 and references therein) are also likely related to the circulation of the boreal water.

The influence of boreal waters on the Mediterranean Tethys might also explain the absence of well-defined positive $\delta^{18}O$ and
$\delta^{13}C$ shifts typically characterizing the EOT in Italian sections, including the GSSP for the EOB (e.g., Houben et al., 2012).
Instead, these sections display distinct negative shifts in $\delta^{18}O$ and $\delta^{13}C$ bulk carbonate values around the EOB (e.g., Jovane et
al., 2007; Brown et al., 2009; Jaramillo-Vogel et al., 2013), resembling the Paratethys records and likely reflecting deep boreal
freshwater incursion through the Paratethys.
**7 Conclusion**
This study provides a comprehensive examination of the EOT and early Oligocene cooling in the eastern Paratethys, with a
focus on the Karaburun composite section. By integrating high-resolution biostratigraphy, geochemistry, sequence
stratigraphy, and precise geochronology, we constructed a robust chronostratigraphic framework spanning the latest Eocene
to early Oligocene. Our findings reveal that the isotopic shifts in $\delta^{18}O$ and $\delta^{13}C$ at Karaburun site align closely with global
records, underscoring the influence of global climatic drivers during this critical transition. However, significant regional
deviations, such as depleted $\delta^{18}O$ values and pronounced stratification, highlight the impact of regional hydrological changes
and basin restriction of the Paratethys. These results emphasize the dual influence of global icehouse dynamics and regional

hydrological processes on the Paratethys during the EOT. The identification of key cooling events, such as the *Glaphyrocysta semitecta* influx for the EOB, and the *Svalbardella* event and its alignment with the Oi-1a glaciation, further enhances the utility of the Karaburun section for refining regional and global stratigraphic correlations. Additionally, the observed sequence stratigraphic patterns illustrate the interplay between eustatic sea-level changes and regional depositional dynamics during this interval. The abrupt depleted values in $\delta^{18}O$ values in benthic foraminifera just after the EOB (ca. 33.7 Ma) in the Paratethys Basin is attributed to boreal water ingression, driven by the closure of the Arctic-Atlantic gateway and the onset of anti-estuarine circulation between the Nordic seas and Atlantic during the EOT. This event, which funneled low-salinity boreal water through the Nordic and North Seas into the Paratethys, aligns with the distribution of boreal taxa like *Svalbardella cooksoniae* and extends as far as the Mediterranean Tethys, explaining isotopic anomalies in these regions. We have to note that the large-scale oceanic circulation scenario proposed here requires the movement of deep waters through narrow and shallow seaways surrounding the Paratethys, such as the Polish Strait, the Alpine Seaway, and smaller Neo-Tethys–Paratethys gateways. However, the timing of their closure and their paleobathymetry during the early Oligocene remain uncertain, as existing paleogeographic maps for this period are inconsistent (e.g. Barrier et al., 2018; Palcu & Krijgsman, 2023; Straume et al., 2024). Therefore, high-resolution paleogeographic reconstructions, integrated with ocean circulation modelling (e.g., Vahlenkamp et al., 2018), are essential for testing and validating this hypothesis. Overall, the Karaburun section emerges as a critical archive for studying the EOT in epicontinental seas. Our findings contribute to a deeper understanding of how global climatic transitions manifest in marginal marine settings and highlight the potential for further high-resolution studies to refine our knowledge of early icehouse climate evolution in the Paratethys region and beyond.

**Author contribution**

MYK designed the study. MYK and SGA conducted fieldwork. TV prepared samples for geochemical analyses. DN performed geochemical analyses. AL conducted geochronological analysis. HB performed the marine palynological analyses. CF analyzed the calcareous nannofossils. MYK wrote the paper with input from all authors. All authors analyzed and discussed the data.

**Competing interests**

The authors declare that they have no conflict of interest.

**Acknowledgements**

This study is funded by TÜBİTAK-2236 fellowship and Horizon 2020 Marie Skłodowska-Curie COFUND action (Project no: 121C058). We thank Pierre Deschamps and Abel Guihou from the CEREGE Envitop analytical facility for their analytical and administrative support. Envitop has received funding from "Excellence Initiative" of Aix Marseille University A*MIDEX - DATCARB project, a French "Investissement d'avenir" program.

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

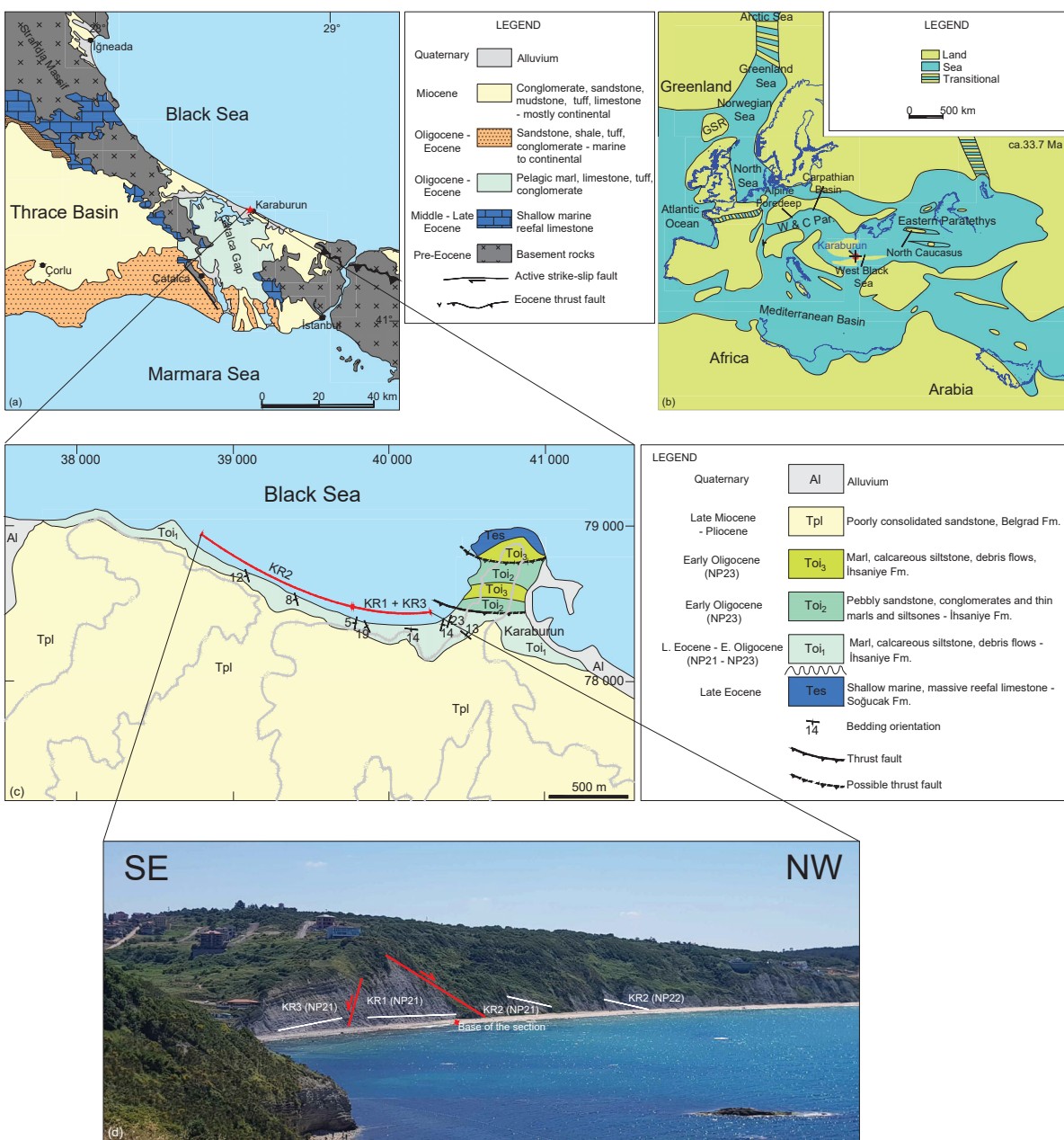

**Figure 1 a) Geological map of the Thrace region (Türkiye) showing the location of the Karaburun area in relation to Thrace Basin, Strandja Massif and Çatalca Gap (modified from Okay et al., 2020). b) Paleogeography of the Paratethys during the early Oligocene (Rupelian, 33.7 Ma) (modified from Sachsenhofer et al., 2018) GSR: Greenland Scotland Ridge. W & C Par.: Western & Central Paratethys. Red dot marks the Karaburun area. c) Geological map of the Karaburun area showing the locations of the sub-sections KR1, KR2 and KR3 (revised from Okay et al., 2019). d) View of the studied sub-sections facing south from Cape Karaburun.**

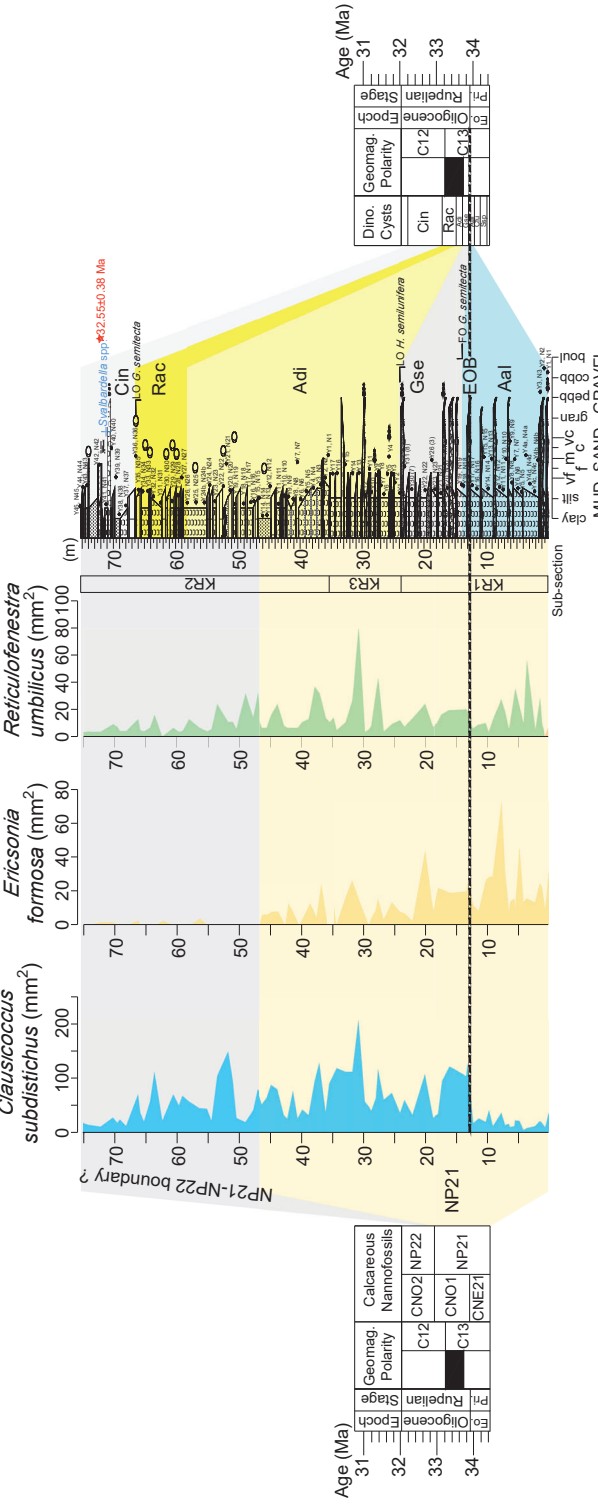

**Figure 2 Stratigraphic log (in meters) of the Karaburun composite section with relative abundances of marker calcareous nannofossils. KR1, KR2, and KR3 are the abbreviations for the studied sub-sections from the Karaburun area. The biostratigraphic (calcareous nannofossil and dinoflagellate cyst) correlations to the geological time scale (Gradstein et al., 2012) are indicated by colored shading. The red star on the log shows the level of the tuff layer (32.65±0.38 Ma) while the blue cross indicates the level of sample with cold-water dinoflagellate *Svalbardella cooksoniae*, indicating a cooling event occurred during the early part of Chron C12r, near the NP21/NP22 boundary.**

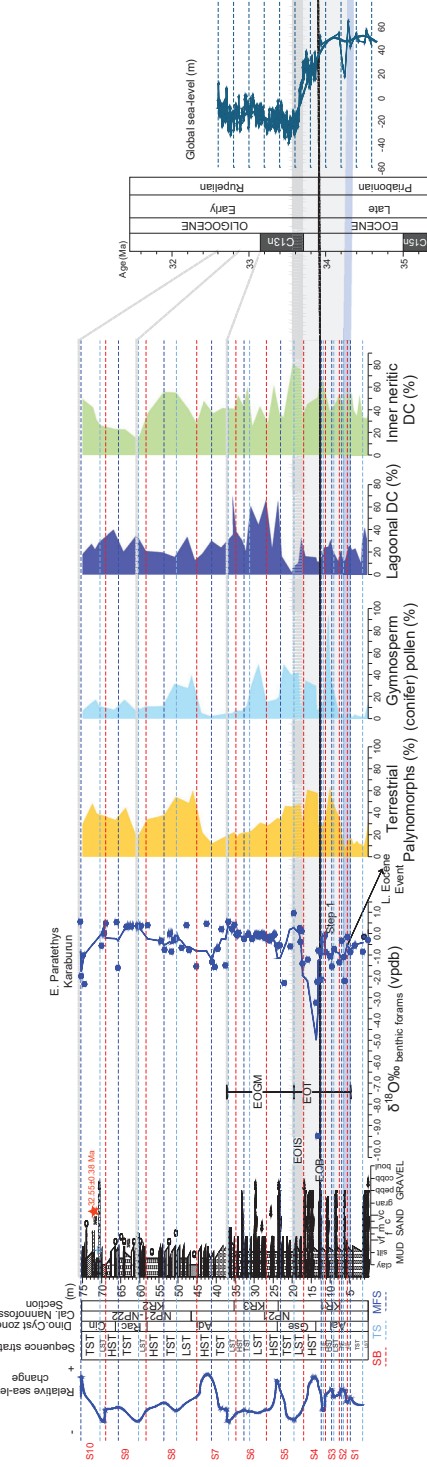

**Figure 3 Stratigraphic log (in meters) of the Karaburun composite section showing sequence stratigraphic interpretations and reconstructed relative sea-level. The sequence stratigraphic interpretation and the reconstructed relative sea-level changes are based on the analysis of benthic foraminifera δ18O values, along with the abundance of terrestrial palynomorphs and lagoonal and inner neritic dinocysts. Red star shows the level of the tuff layer on the log while blue cross indicates the level of sample with cold-water dinocyst *Svalbardella cooksoniae*. Correlation to the reconstructed global global sea-level curve (Miller et al., 2020) and to the geological time scale (Gradstein et al., 2012) could be seen on the right side. S: Sequence, SB: Sequence boundary, TS: Transgressive surface, MFS: Maximum flooding surface, LST: Lowstand systems tract, TST: Transgressive Systemes Tract. HTS: Highstand Systems Tract, EOB: Eocene Oligocene Boundary, EOT: Eocene Oligocene Transition, EOIS: Earliest Oligocene oxygen isotope step, EOGM: Early Oligocene glacial maximum.**

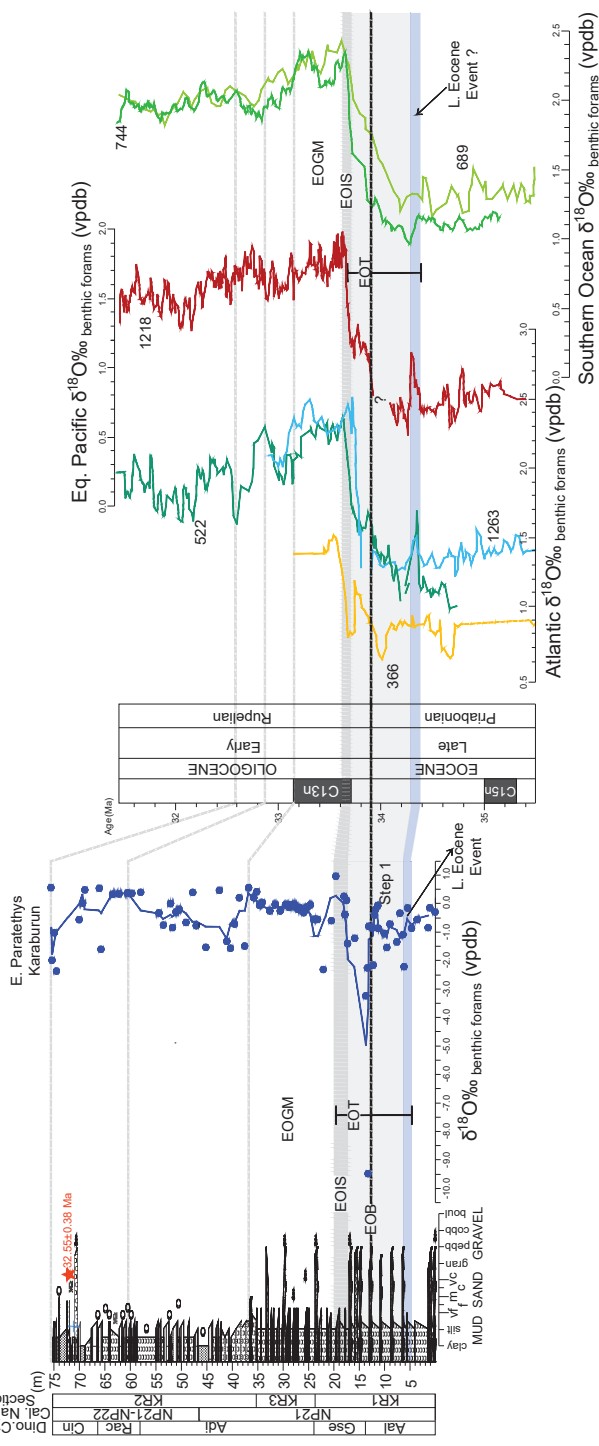

**Figure 4** The stratigraphic log (in meters) of the Karaburun composite section including the results of benthic foraminifera δ¹⁸O record (blue dots and line showing three-point running mean), highlighting the chronostratigraphic characteristics of the Eocene-Oligocene Transition (EOT). Apparent correlations were established by aligning the Karaburun data with high-resolution deep sea benthic foraminifera δ¹⁸O records from the South Atlantics sites 522 (Zachos et al., 1996) and 1263 (Langton et al., 2016), and compared to the tropical Atlantic site 366 (Langton et al., 2016), the Southern Ocean sites 744 and 689 (Zachos et al., 1996; Diester-Haass and Zahn, 1996), and the equatorial Pacific site 1218 (Coxall and Wilson, 2011). Key features in the δ¹⁸O records, such as positive and negative shifts and their amplitudes, were used to define EOT characteristics, including the Late Eocene Event, the Earliest Oligocene Oxygen Isotope Step (EOIS) and the Early Oligocene Glacial Maximum (EOGM). The Eocene-Oligocene Boundary (EOB) was identified through biostratigraphic analyses (see section 5.1). The red star on the log marks the tuff layer, while the blue cross indicates the occurrence of the cold-water dinocyst *Svalbardella cooksoniae*.

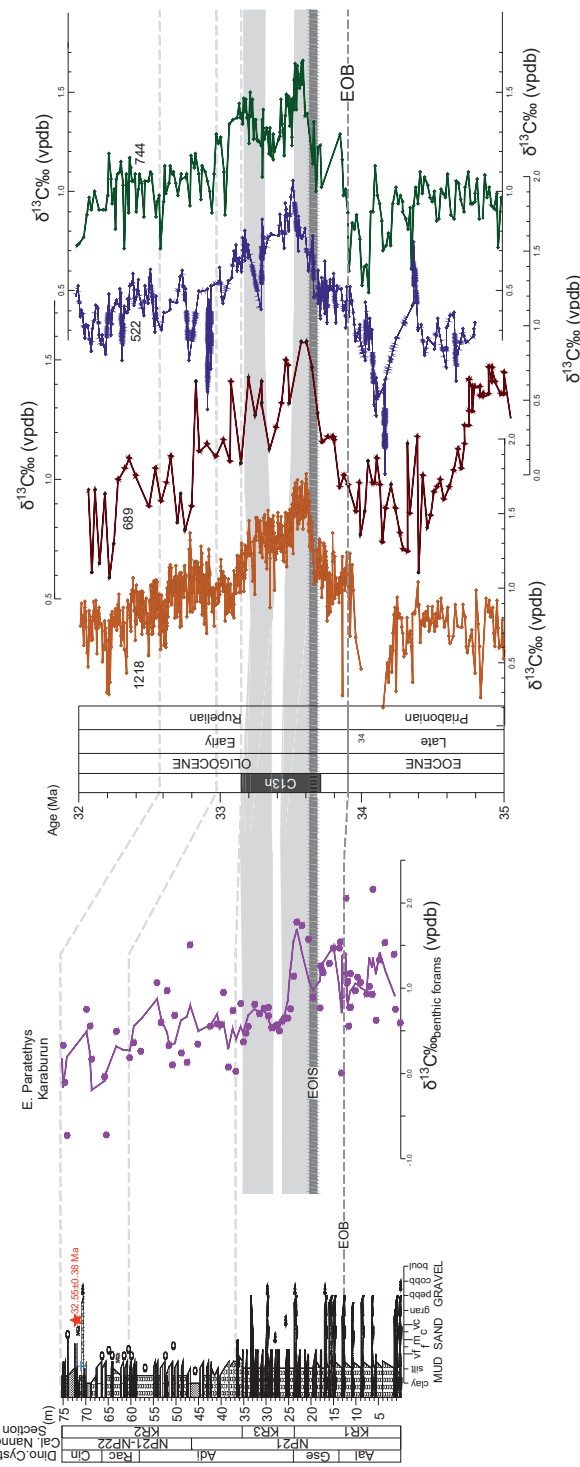

**Figure 5 Benthic foraminifera δ¹³C record (pink dots and line showing three-point running mean) of the Karaburun composite section along the stratigraphic log (in meters) highlighting the chronostratigraphic features of the Eocene-Oligocene Transition (EOT). Correlations were made with deep marine records from Site 689 in the sub-Antarctic Atlantic (Diester-Haass and Zahn, 1996), Site 1218 in the equatorial Pacific (Coxall and Wilson, 2011), Site 522 in the South Atlantic (Zachos et al., 1996), and Site 744 in the Southern Ocean (Zachos et al., 1996). Alignment of high-resolution benthic foraminifera δ¹³C records from these sites with the Karaburun data revealed corresponding double peak isotope feature (gray shading) after the Earliest Oligocene Oxygen Isotope Step (EOIS) within the Early Oligocene Glacial Maximum (EOGM). The Eocene-Oligocene Boundary (EOB) was determined through biostratigraphic analysis (see section 5.1). The red star marks the tuff layer, while the blue cross indicates the level of sample containing the cold-water dinocyst *Svalbardella cooksoniae*.**

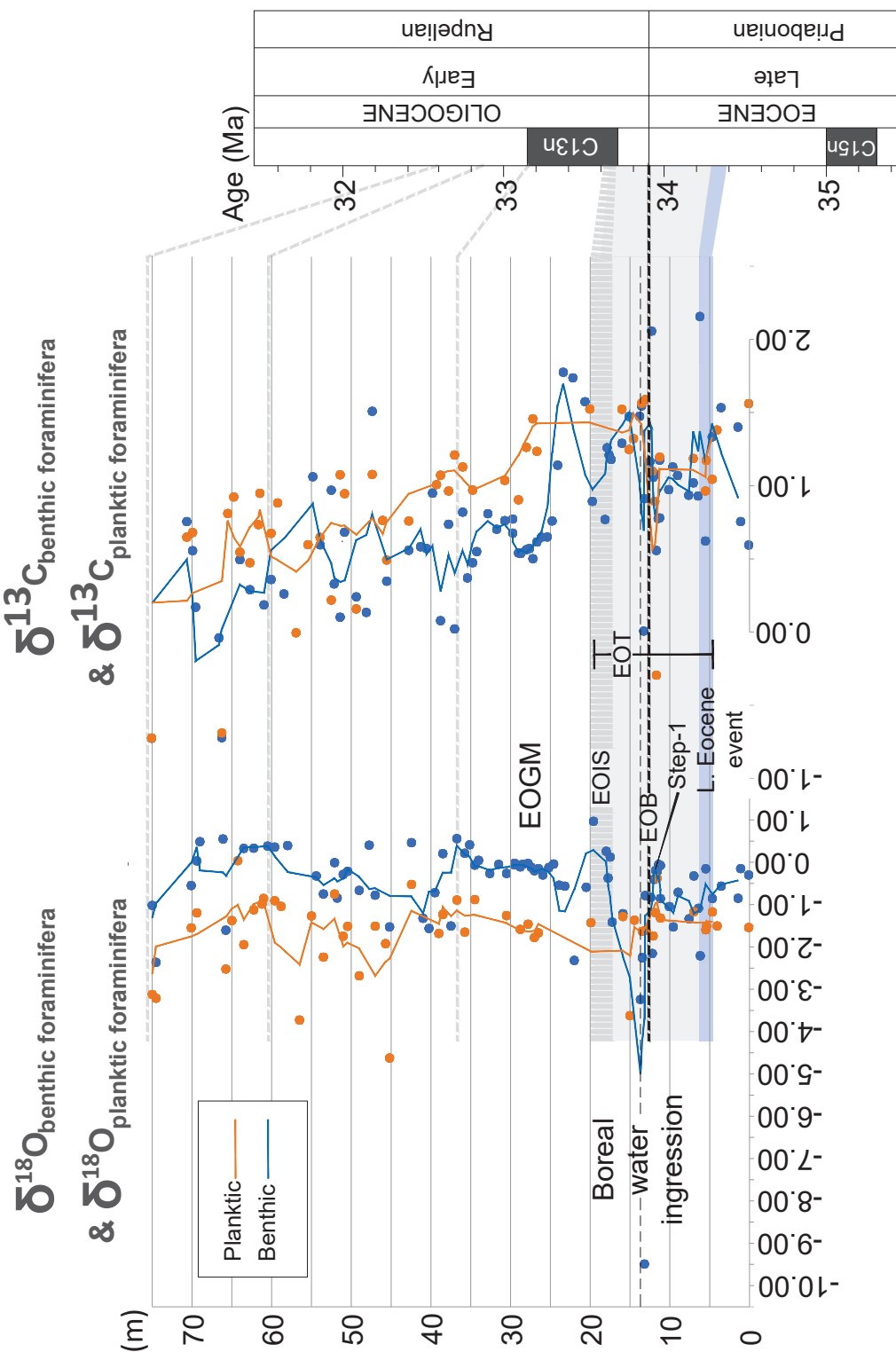

**Figure 6 Oxygen (δ¹⁸O) and Carbon (δ¹³C) stable isotope values for the benthic and planktic foraminifera from the Karaburun composite section. The black dashed line represents the level for the boreal water ingression just after the Eocene-Oligocene Boundary (EOB). EOT: Eocene-Oligocene Transition. EOIS: Earliest Oligocene Oxygen Isotope Step. EOGM: Early Oligocene Glacial Maximum.**

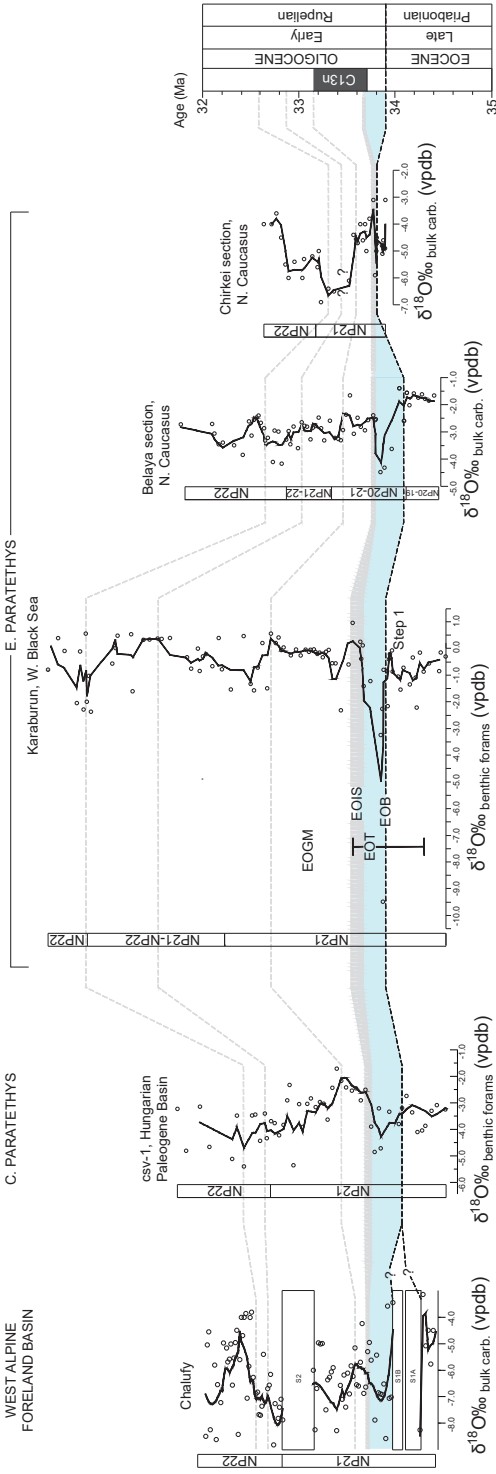

**Figure 7** Oxygen stable isotope (δ18O) values and the characteristic negative δ18O shift during the EOT observed in various Paratethys marine records. Correlations highlight the interval (light blue shading) between the Eocene-Oligocene Boundary (EOB) and the Earliest Oligocene Oxygen Isotope Step (EOIS, gray dashed line) across various sections, including the Chalufy Section in the West Alpine Foreland Basin, which links the Western Paratethys to the Mediterranean Tethys (Soutter et al., 2022); the csv-1 core from the Hungarian Paleogene Basin (Ozsvárt et al., 2016); the Karaburun Section (this study); and the Belaya (van der Boon et al., 2019) and Chirkei (Gavrilov et al., 2017) sections from the Northern Caucasus. To account for varying sample resolutions, a three-point running mean filter was applied uniformly to all sites.

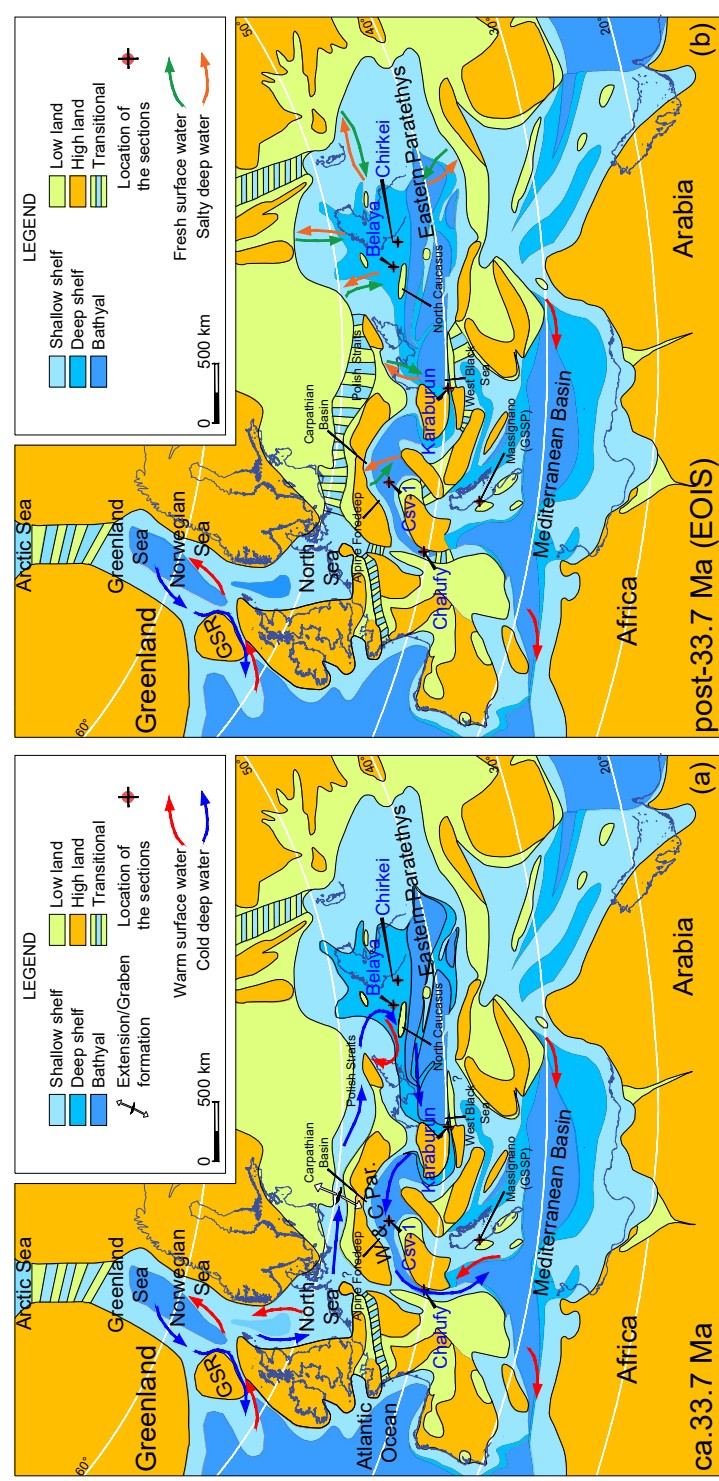

**Figure 8** Paleogeographic maps at 33.7 Ma and post-33.7 Ma during the EOIS (The Earliest Oligocene Oxygen Isotope Step) (modified from Sachsenhofer et al., 2018). (a) boreal water anti-estuarine circulation at ca. 33.7 Ma through the Paratethys, extending into the Mediterranean Tethys. GSR: Greenland Scotland Ridge. (b) estuarine-circulation in the Paratethys during the EOIS due to the geographic restrictions (closure of the Mediterranean and other seaways due to the major sea-level fall) and hydrological changes (increased fresh water influx from land).