# Peer review of "The Eocene-Oligocene Transition in the Paratethys: Boreal Water 2 Ingression and its Paleoceanographic Implications"

_EGUsphere, 2025_

## Author Response (AR1)

Dear editor & dear reviewers,

Please consider our revised manuscript. We have implemented the suggestions of the editor and reviewers and also commented below on each of the points raised. As suggested by the editor, we have now further elaborated on the potential causes for the shift in circulation regime from anti-estuarine to estuarine conditions.

We thank the editor and both reviewers for the opportunity to revise our paper and for the extremely helpful comments. With all the suggested and required additions, clarifications and corrections our revised manuscript is now much clearer and more concise.

**Responses to the reviewer 1:**

**1. Even in the North Sea area microfossil biostratigraphy is suffering from endemism making correlations with the international zonation sometimes difficult. Is the not a fortiori an issue in the Paratethys area ?**

Thank you for your comment. We acknowledge that endemism can pose challenges for biostratigraphic correlations, particularly in semi-isolated basins like the Paratethys. However, in the context of our study, this issue becomes more prominent starting from the NP23 Calcareous Nannofossil zonation, which was not covered here. During NP21 and (early part of) NP22, the Paratethys remained connected to the world oceans, as also evidenced by our findings. This connection is reflected in the presence of microfossils—including dinoflagellates, calcareous nannofossils, and both benthic and planktic foraminifera—that were instrumental in constructing the international biozonation schemes.

Moreover, the regional biozonations can still be effectively correlated with international schemes through integrated stratigraphic approaches, such as combining biostratigraphy with chemostratigraphy. In our study, we have carefully considered these factors and have utilized the most robust correlations available to ensure the reliability of our stratigraphic framework.

**2. use consistent sedimentologic terminology : hemipelagic ( line 115) and also bathyal (line 288) together with so many indications of shallow water and lagoonal microfossils, alluvial fans...**

Thank you for highlighting the importance of consistency. We will carefully review and improve the consistency of sedimentological terminology.

We use the term *hemipelagic* to describe sediments deposited on a continental shelf, consisting primarily of clay- and silt-sized terrigenous grains, with additional biogenic material derived either from the nearest landmass or from organisms living in the water column. Given this definition, we prefer to retain the term as it is (Line 115).

Regarding the depositional environment interpretation, we acknowledge that the original sentence was not comprehensive in addressing shallow-water indicators. To improve clarity, we have revised it as follows:

**Original:**
*"Based on lithological, sedimentological, and paleontological characteristics, we interpret the depositional environment as a bathyal outer-shelf setting."*

**Revised:**
*"Based on lithological, sedimentological, and paleontological characteristics, we interpret the depositional environment as a shelf setting exhibiting littoral and neritic characteristics, depending on fluctuating sea levels."*

**3. would a word of explanation about the ratio of 32 examined zircons versus only 11 usable zircons ( line 183) not helping to confirm the value of the new tuff age ? ( the additional table S3 is not helping in this respect).**

Thank you for your suggestion. We acknowledge that the discrepancy between the 35 examined zircons and the 11 usable zircons may require further clarification. We have added a brief explanation in the text to clarify the selection criteria for usable zircons as below;

*"We applied rigorous filtering based on zircon morphology, U-Pb concordance, and common Pb content to ensure the reliability of the final age determination. The dataset was filtered for concordant grains using the Concordia distance method of Vermeesch (2021), which applies isometric logratios with a discordance threshold of 10 (SI units) and a reverse discordance threshold of 5 (SI units)."*

**4. the absence of microfossils on itself is generally not a good argument for a biostrat zonation ( line 206) and the term 'bizarre' dinoflagellate ( line 280) is not helping to understand its biostrat value.**

We respectfully disagree with Referee 1's comment. The extinction event of the rosette-shaped *Discoaster* species is a well-established marker for defining the base of Zone NP21. Furthermore, this bioevent is not an isolated criterion but is corroborated by its correlation with dinoflagellate cyst zonations and geochemical data (chemostratigraphy), further reinforcing its reliability.

We have removed the word "bizarre" as suggested by the Referee 1.

**5. Regarding the sequence stratigraphic interpretations (section 5.3) :**

**I prefer the term 'relative sea-level evolution' as is used in the figures. The comparison with global SL is hard to do in this case as 10 cycles are identified in only less than 2 million years...Also it can be expected that vertical tectonic influence plays an important role in the delivery of detrital sediments (a.o. the alluvial fans) to the basin seen the Paratethys context ...**

We agree with the comment and will use "relative sea-level evolution" instead of "relative sea-level fluctuations" throughout the manuscript. The necessary changes have been made accordingly.

The comparison with global sea-level reconstructions can be effectively conducted by comparing the Eastern Paratethys relative sea-level evolution with the smoothed global curve of Miller et al. (2020). While tectonic influence is certainly expected, the parallel trends observed between the Eastern Paratethys, the Mediterranean (Italian sections), and the North Sea (see Section 6.3) further support the dominance of an eustatic control.

**I do not understand the reference to Westerhold et al. 2020 relating thickness to orbital changes .**

**Original:**

*"This variation in depositional thickness may be linked to orbital forcing across the late Eocene to early Oligocene (e.g., Westerhold et al., 2020)."*

We agree that the original sentence was unclear and we have revised it for better clarity. Specifically, we have elaborated on the role of obliquity forcing in controlling depositional thickness during the latest Eocene. This orbital influence has been recognized in both

terrestrial and marine records, as supported by previous studies (e.g., Abels et al., 2011; Jovane et al., 2006; Boulila et al., 2021). The revised sentence is as below:

**Revised:**

*"This variation in depositional thickness could be related to obliquity forcing during the latest Eocene, a process that has been identified in several terrestrial and marine records (e.g., Abels et al., 2011; Jovane et al., 2006; Boulila et al., 2021)."*

**Therefore I would consider the detailed interpreted Relative Sea Level curve derived in the section as a well documented RSL to compare in the future to other such RSL curves to be constructed in the area rather than already pretending now its general character in the area.**

**Original:**

*"These high resolution sea-level fluctuations provide a refined reconstruction of Eastern Paratethys sea-level changes, improving upon previous studies (e.g., Popov et al., 2010)."*

The original sentence has been changed as below as suggested by the Referee 1.

**Revised:**

*"These high-resolution sea-level fluctuations offer a refined reconstruction of Eastern Paratethys sea-level changes (e.g., Popov et al., 2010) and provide a valuable framework for future studies to further reveal the history of Paratethys sea-level evolution."*

**Responses to the reviewer 2:**

**Line 508: The interpretation of observed geochemical signatures as local effects appears contradictory given the reported basin-wide distribution of these features. Could the authors reconcile this apparent discrepancy between localized interpretations and regional-scale observations?**

*We acknowledge Reviewer 2's concern regarding the contradiction between interpreting these features as "local effects" and their observed basin-wide distribution. To resolve this inconsistency, we have revised the relevant sentences by replacing "local" with "regional" to better reflect the broader spatial extent of these geochemical signatures.*

**Line 519: The proposed linkage between Atlantic-derived westerlies and enhanced precipitation warrants further mechanistic explanation. Specifically, does you mean intensification of vapor transport through strengthened zonal winds, or altered storm track patterns associated with EOT cooling?**

*Thank you for raising this point. We link the enhanced precipitation to the intensification of the westerlies and the reorganization of oceanic and atmospheric circulation (e.g., Hou et al., 2022). As global cooling progressed and the meridional temperature gradient steepened, the westerlies strengthened, increasing eastward moisture transport. Intensified westerlies likely enhanced vapor transport from the Atlantic into Eurasia, contributing to higher precipitation in the western and central Paratethys (e.g., Koscis et al., 2014; Li et al., 2018).*

*We have incorporated these explanations into the manuscript as below;*

*"During the Rupelian (35–31 Ma), the dominant influence of Atlantic derived westerlies likely brought increased precipitation with a depleted $\delta^{18}O$ signature to the western and central Paratethys. This interpretation was supported by $\delta^{18}O_{PO_4}$ values from herbivore tooth enamel, which reflect the depleted isotopic composition of drinking water (Kocsis et al., 2014). The*

*enhanced precipitation was likely due to the intensification of the westerlies and the reorganization of oceanic and atmospheric circulation (e.g., Hou et al., 2022). As global cooling progressed and the meridional temperature gradient steepened, the westerlies strengthened, enhancing vapor transport from the Atlantic into Eurasia, contributing to higher precipitation in the western and central Paratethys (e.g., Koscis et al., 2014; Li et al., 2018). Modeling studies further suggest prevailing westerly winds during winter at mid-high latitudes in the Rupelian (Li et al., 2018)."*

**Lines 524-526 & 543: The assertion of enhanced evaporation during cooling appears counterintuitive given reduced latent heat transfer capacity at lower temperatures. Could this paradox be resolved through 1) basin geometry changes altering surface area/volume ratios, or 2) wind-driven evaporation effects overriding thermal constraints?**

*We appreciate Reviewer 2's insightful question, which indeed requires clarification. Both factors mentioned—the influence of basin geometry and wind-driven evaporation—likely contributed to the more evaporitic conditions in the Eastern Paratethys compared to the Western Paratethys.*

*We have clarified these points in the revised manuscript as below;*

*"Basin geometry, particularly water depth, plays a crucial role in evaporation dynamics within a semi-isolated system like the Paratethys. The Western Paratethys, characterized by deeper basins, had a higher heat capacity and greater thermal inertia, leading to moderated surface temperatures and lower evaporation rates. In contrast, the shallower Eastern Paratethys had lower heat capacity, allowing for rapid surface warming, especially in warmer seasons, which enhanced evaporation. Additionally, wind-driven effects likely played a significant role. In deeper basins, wind-driven mixing redistributed heat within the water column, reducing extreme surface warming and stabilizing evaporation rates. In shallower areas, reduced mixing allowed surface waters to remain warmer, leading to increased evaporation, particularly under strong seasonal winds." Hence, further east, evaporation over the shallower Eastern Paratethys may have added moisture to westerly air trajectories, resulting in relatively less negative δ¹⁸O values in the Northern Caucasus (Karaburun, Belaya and Chirkei sections in Figure 7) and increased inland precipitation (Figure 7)."*

**Lines 565 & 586: The multiple references to precipitation increases would benefit from explicit differentiation between moisture sources (e.g., Mediterranean versus boreal origins) and forcing mechanisms (orographic effects vs. atmospheric circulation changes).**

*As mentioned in our penultimate response, we have further clarified the multiple references to precipitation increases by explicitly differentiating between moisture sources (e.g., Atlantic-sourced westerlies, as in Koscis et al., 2014) and forcing mechanisms (e.g., reorganization of oceanic and atmospheric circulation, as in Hou et al., 2022). These aspects have been addressed in the revised manuscript to improve clarity and provide a more detailed interpretation.*

**Lines 626-627: The authors describe the changes of ocean currents in the Paratethys and thus the inferred strengthening of Atlantic Meridional Overturning Circulation (AMOC) requires clearer justification.**

*We agree that this point could have been explained more clearly in the manuscript. We have refined the explanation accordingly in the revised version as below;*

*"Together, the Atlantic-Arctic closure and onset of anti-estuarine circulation events could have triggered or intensified the Atlantic Meridional Overturning Circulation (AMOC) (e.g., Abelson and Erez, 2017; Coxall et al., 2018; Hutchinson et al., 2019). Our data also indicate the onset of anti-estuarine circulation in the Paratethys around 33.7 Ma, coinciding with the development of similar circulation patterns between the Nordic Seas and the North Atlantic (Abelson and Erez, 2017). We propose a hypothetical pathway for this circulation: warm surface water entering from the North Sea would have flowed into the eastern Nordic Seas, joining the warm surface waters from the North Atlantic. After losing heat, this water likely sank in the northern Nordic basin, flowed southward as deep water, and eventually reached the North Atlantic, the North Sea, the Paratethys, and the Mediterranean Tethys, forming an interhemispheric northern-sourced circulation cell.*

*The onset of anti-estuarine circulation in the North Sea was likely related to the deepening of the Greenland-Scotland Ridge (GSR), similar to the development of anti-estuarine circulation in the Nordic Seas. During most of the Paleogene, the GSR was shallower, restricting deep-water exchange between the North Atlantic and the Nordic Seas. A key feature of the modern AMOC is the formation of North Atlantic Deep Water, which flows over the GSR from the Nordic Seas. The initiation of Nordic anti-estuarine circulation events around the EOT likely enhanced deep-water formation, strengthening the Atlantic Meridional Overturning Circulation (AMOC) and establishing an interhemispheric northern-sourced circulation cell (e.g., Abelson and Erez, 2017; Coxall et al., 2018; Hutchinson et al., 2019). This suggests that by ~33.7 Ma, the Paratethys, with its anti-estuarine circulation, was integrated into this larger circulation system, contributing to global ocean circulation."*

**Line 634: the deepening of the GSR facilitated increased exchange between the Atlantic and Nordic Seas, enabling the salinization of North Atlantic surface waters. Why is the Nordic Sea more salty than the Arctic?**

*Geological reconstructions of the Barents Sea suggest that it was the only viable connection between the Arctic and the Atlantic (including the Nordic Seas) during the late Eocene. Strong evidence indicates that the Arctic underwent freshening during this period. The closure of its connection with the Atlantic—due to the emergence of a land bridge in the Barents Sea region (e.g., Hutchinson et al., 2019)—isolated the Arctic, preventing saltwater inflow.*

*Meanwhile, the Nordic Seas maintained a shallow connection with the North Atlantic through the Greenland-Scotland Ridge (GSR). This connectivity allowed for greater exchange with the saltier North Atlantic, leading to relatively higher salinity in the Nordic Seas compared to the Arctic, which remained fresher due to its isolation.*

**Lines 648-650: an estuarine circulation during the EOT, does this mean AMOC weaken accordingly, and why? Moreover, the authors indicate an anti-estuarine circulation occurs at ca. 33.7 Ma. Both seem to happen at about the same time. The temporal coincidence of estuarine and anti-estuarine circulation modes raises questions about their spatial configuration.**

*The occurrence of anti-estuarine and estuarine circulation does not coincide temporally. The anti-estuarine circulation occurs around 33.7 Ma, while estuarine circulation begins shortly after the EOIS, as shown in Figure 6. The transition observed in the Paratethys is likely linked to major sea-level changes (major fall during the EOIS), which restricted water exchange between the North Sea and the Paratethys basins. However, the anti-estuarine circulation between the Nordic Seas and the North Atlantic continues throughout this period. Therefore, the shift to estuarine circulation in the Paratethys does not imply a weakening of the AMOC.*

**Figure 8: pre- and post-33.7 Ma characteristics of surface and deep waters would significantly strengthen the visualization of circulation regime changes.**

*We appreciate this suggestion and agree that adding a comparison of pre- and post-33.7 Ma characteristics of surface and deep waters would enhance the visualization of circulation regime changes. We have now modified Figure 8 accordingly and included an additional figure to improve the illustration of these changes.*